# Stagnant ice and age modelling in the Dome C region, Antarctica

**Ailsa Chung[1], Frédéric Parrenin[1], Daniel Steinhage[2], Robert Mulvaney[3], Carlos Martín[3], Marie G. P. Cavitte[4], David A. Lilien[5,6], Veit Helm[2], Drew Taylor[7], Prasad Gogineni[7], Catherine Ritz[1], Massimo Frezzotti[10], Charles O'Neill[8], Heinrich Miller[2], Dorthe Dahl-Jensen[5,6], and Olaf Eisen[2,9]**

[1]Univ. Grenoble Alpes, CNRS, INRAE, IRD, Grenoble INP, IGE, 38000 Grenoble, France
[2]Glaciology, Alfred-Wegener-Institut Helmholtz-Zentrum für Polar-und Meeresforschung, Bremerhaven, Germany
[3]British Antarctic Survey, Cambridge, UK
[4]Earth and Life Institute (ELI), Université catholique de Louvain-La-Neuve (UCLouvain), Louvain-la-Neuve, Belgium
[5]Physics of Ice, Climate and Earth, Niels Bohr Institute, University of Copenhagen, Copenhagen, Denmark
[6]Centre for Earth Observation Science, University of Manitoba, Winnipeg, Canada
[7]Remote Sensing Center, University of Alabama, Tuscaloosa, AL, USA
[8]EH Group, Tuscaloosa, AL, USA
[9]Fachbereich Geowissenschaften, Universität Bremen, Bremen, Germany
[10]Department of Science, University Roma Tre, Rome, Italy

**Correspondence:** Ailsa Chung (ailsa.chung@univ-grenoble-alpes.fr) and Frédéric Parrenin (frederic.parrenin@univ-grenoble-alpes.fr)

**Abstract.** The European Beyond EPICA project aims to extract a continuous ice core of up to 1.5 Ma, with a maximum age density of $20 \, \mathrm{kyr \, m^{-1}}$ at Little Dome C (LDC). We present a 1D numerical model which calculates the age of the ice around Dome C. The model inverts for basal conditions and accounts either for melting or for a layer of stagnant ice above the bedrock. It is constrained by internal reflecting horizons traced in radargrams and dated using the EPICA Dome C (EDC) ice core age profile. We used three different radar datasets ranging from a $10\,000 \, \mathrm{km^2}$ airborne survey down to 5 km long ground-based radar transects over LDC. We find that stagnant ice exists in many places, including above the LDC relief where the new Beyond EPICA drill site (BELDC) is located. The modelled thickness of this layer of stagnant ice roughly corresponds to the thickness of the basal unit observed in one of the radar surveys and in the autonomous phase-sensitive radio-echo sounder (ApRES) dataset. At BELDC, the modelled stagnant ice thickness is $198 \pm 44 \, \mathrm{m}$ and the modelled oldest age of ice is $1.45 \pm 0.16 \, \mathrm{Ma}$ at a depth of $2494 \pm 30 \, \mathrm{m}$. This is very similar to all sites situated on the LDC relief, including that of the Million Year Ice Core project being conducted by the Australian Antarctic Division. The model was also applied to radar data in the area 10–15 km north of EDC (North Patch), where we find either a thin layer of stagnant ice (generally $< 60 \, \mathrm{m}$) or a negligible melt rate ($< 0.1 \, \mathrm{mm \, yr^{-1}}$). The modelled maximum age at North Patch is over 2 Ma in most places, with ice at 1.5 Ma having a resolution of 9–$12 \, \mathrm{kyr \, m^{-1}}$, making it an exciting prospect for a future Oldest Ice drill site.

## 1 Introduction

The Mid-Pleistocene Transition (MPT) marks the change in climate glacial-interglacial cycles from those with low amplitude and around 41 kyr periodicity to the current high-amplitude cycles of 100 kyr on average (Clark et al., 2006). The MPT took place between 1250 and 700 ka. Understanding the factors that affected past glacial cycles can help us to construct models to predict how human emissions will affect the climate in the future. Ice cores provide unique insights into the climate of the past (Petit et al., 1999; EPICA members, 2004; Ruth et al., 2007; Dome Fuji Ice Core Project Members, 2017). To date, the oldest continuous ice core record extracted from Antarctica is from the EPICA Dome C

(EDC) core, which goes back $\sim 800$ ka (Bazin et al., 2013). The Oldest Ice challenge defined by the International Partnership in Ice Core Sciences (IPICS) community now aims to study the MPT using an ice archive from Antarctica. Potential sites have been defined $\sim 35$ km southwest of EDC, near a secondary dome called Little Dome C (LDC). There, the mountainous bedrock is thought to prevent basal melting (Parrenin et al., 2017; Passalacqua et al., 2017; Lilien et al., 2021), and the overall glaciological conditions satisfy the requirements recommended by the IPICS community (Fischer et al., 2013; Van Liefferinge and Pattyn, 2013). The European Beyond EPICA project aims to extract a continuous ice core of up to 1.5 Ma, with a maximum age density of 20 kyr m$^{-1}$ at this site called Beyond EPICA Little Dome C (BELDC). The Australian Antarctic Division (AAD) has also selected a drill site at LDC 5 km from BELDC for their Million Year Ice Core (MYIC) project.

A fundamental question for the Oldest Ice sites on the East Antarctic Plateau such as BELDC and MYIC is whether the deepest ice lying just above the bedrock proves useful for palaeoclimate reconstructions. When examining the Vostok ice core, a layer of deformed ice was found just above the bedrock (Souchez et al., 2002). Isotopic analysis showed that the bottom 228 m of the ice core had deformed as a result of folding and intermixing at a sub-metre scale. Generally known as the basal layer, the mechanics of this deformed ice are not well understood. At the bottom of the EDC ice core, there is a section of around 60–70 m where Tison et al. (2015) found that the palaeoclimatic signal had been disturbed, perhaps due to a chemical sorting mechanism cause by the ice being close to the melting point. While the isotopic composition of this ice was studied by Tison et al. (2015), the interpretation of these results remains challenging. The mechanical stress on the deepest ice has distorted the timescale and left no continuous record. Basal ice was difficult to observe using previous radar systems due to the presence of an echo-free zone (EFZ). The EFZ was caused by a loss of power in the deepest ice, which is close to the melting point. The effect is sometimes still observed where the related change in the dielectric properties could be due to folding, buckling, recirculation, recrystallisation or a sharp thermal transition (Drews et al., 2009). The EFZ is often a band just above the bed and up to hundreds of metres thick, where there are no observed layers in radargrams. However, during the analysis of the EPICA Dronning Maud Land ice core (EDML), Ruth et al. (2007) found stratigraphically conserved ice at the EDML in the upper 50 % of the EFZ. Therefore, even with the absence of visible layers in the radargram, there can still be a detectable palaeoclimatic signal in the ice.

In their modelling work, Parrenin et al. (2007) used the ice–bedrock interface as the bottom boundary condition. But given the smoothness of the basal isochrones compared to the roughness of the observed ice–bedrock interface, it was suggested that a layer of stagnant ice could exist above some areas of the bedrock (Parrenin et al., 2017). More recently, a basal layer of ice with different dielectric properties, called the "basal unit", has been documented in radar imaging. Cavitte (2017) showed that this basal unit was prevalent over the LDC subglacial topography. Lilien et al. (2021) found that at the Beyond EPICA drill site on Little Dome C in Antarctica the basal unit is around 200 m thick. They found that at this depth there was a change in return power and lateral coherency of the signals, which is likely to come from a change in the physical properties of the ice. This is in contrast to the EFZ, which seems to be caused by a backscatter power that is sufficiently far below the noise level and therefore cannot be imaged (Drews et al., 2009). Lilien et al. (2021) assessed the age–depth profile at the BELDC site, inverting the optimal value of the thickness of a layer of stagnant ice, which was found to be close to the observed thickness of the basal unit. Here, we develop the idea further by investigating the whole Dome C region using a similar model to Lilien et al. (2021) but with a different numerical scheme and optimisation method (see the Supplement). This approach will elucidate the spatial extent of this inferred stagnant ice layer and its impact on the age profile in the region.

In this study, measurements made with an autonomous phase-sensitive radio-echo sounder (ApRES) show that the vertical velocity profile at LDC is better described when there is a layer of stagnant ice above the bedrock (Fig. 3). We also show observations of a basal unit in radar surveys, which could provide further evidence of a stagnant ice basal layer. We present a 1D numerical model which uses inverse methods to infer a layer of stagnant ice from the isochronal information and assess its relevance over a non-stagnant ice model. This model is applied to three radar datasets acquired in recent years, which cover the larger Dome C area. We then compare the thickness of the modelled stagnant ice with the basal unit observed in the radar and propose that they are one and the same. We discuss the expectations for the Beyond EPICA ice core based on our modelling results and also take a closer look at an area 10–15 km north of EDC, known as North Patch (NP), which could be an interesting site for future Oldest Ice drilling projects.

## 2 Numerical age model

### 2.1 Method summary

We use a modified version of the Parrenin et al. (2017) 1D numerical age model. The model is applicable to areas centred on domes or near ice divides where there is little to no horizontal ice flow and the bedrock relief is relatively smooth. It is based on the pseudo-steady assumption, which means that the geometry and vertical velocity profile are steady state. The only temporal variation is related to accumulation and derived from the EDC ice core reconstruction (Bazin et al., 2013). The modifications we apply to the Parrenin et al. (2017) model relate to the thermal representation.

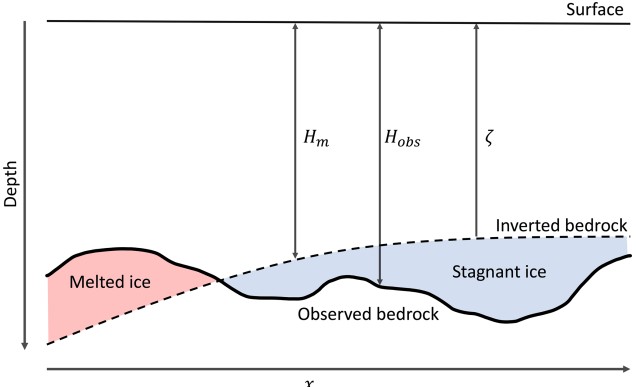

**Figure 1.** Diagram showing mechanical ice thickness $H_{\mathrm{m}}$ and radar-observed ice thickness $H_{\mathrm{obs}}$. The melted ice case is in red and stagnant ice case is in blue. The normalised ice elevation $\zeta$ is 0 at the inverted bedrock $H_{\mathrm{m}}$ and 1 at the surface.

While Parrenin et al. (2017) include an explicit temperature scheme, accounting for the geothermal flux, we use a simpler approach which allows for a layer of "stagnant ice" above the bedrock. We do so by using a "mechanical ice thickness" $H_{\mathrm{m}}$, which is different to the observed ice thickness $H_{\mathrm{obs}}$ (Fig. 1). We also assume that there is no melting at $H_{\mathrm{m}}$. $H_{\mathrm{m}} < H_{\mathrm{obs}}$ (stagnant ice case) means there is a layer of stagnant ice of thickness $H_{\mathrm{obs}} - H_{\mathrm{m}}$, and the age goes to infinity at depth $H_{\mathrm{m}}$. We label this ice as "stagnant" as the best-fit thinning profile of the 1D model does not pass below $H_{\mathrm{m}}$, though from observations we can see that ice continues to depth $H_{\mathrm{obs}}$, so we infer a vertical ice flow velocity of 0 for this layer. $H_{\mathrm{m}} > H_{\mathrm{obs}}$ (melting case) means that there is some melting at $H_{\mathrm{obs}}$, which is calculated by the truncation of the vertical velocity profile at $H_{\mathrm{obs}}$. In this case, the age profile is truncated to $H_{\mathrm{obs}}$, meaning that the age at $H_{\mathrm{obs}}$ is finite.

## 2.2 Forward model

As in Parrenin et al. (2017), we compute the true age from a so-called steady age using the change of time variable

$$\bar{t} = \int_0^t r(t')\mathrm{d}t', \tag{1}$$

where $r(t)$ is the ratio of the EDC accumulation rate (Bazin et al., 2013) to its temporally averaged value over the last 800 ka (Fig. 2 of Parrenin et al., 2017). Beyond the extent of the EDC record (800 ka), we assume $r(t) = 1$.

The steady age of the ice ($\bar{\chi}$) at spatial position $x$ is given by

$$\bar{\chi}(\zeta) = \int \frac{H_{\mathrm{m}}}{\bar{a}\tau(\zeta)}\mathrm{d}\zeta, \tag{2}$$

where $\bar{a}$ is the average accumulation rate, and $\zeta$ is the normalised vertical coordinate (0 at $H_{\mathrm{m}}$, 1 at the surface).

Since there is no melting at $H_{\mathrm{m}}$, the thinning function $\tau$ is equal to the horizontal flux shape function $\omega$. Here we use the Lliboutry velocity profile (Lliboutry, 1979)

$$\tau(\zeta) = \omega(\zeta) = 1 - \frac{p+2}{p+1}(1-\zeta) + \frac{1}{p+1}(1-\zeta)^{p+2}, \tag{3}$$

where the parameter $p$ modifies the non-linearity of $\omega$ (with larger $p$, $\omega$ becomes more linear). In the melting case, the basal melt rate is calculated by $m = \bar{a}\omega(H_{\mathrm{obs}}/H_{\mathrm{m}})$.

## 2.3 Optimisation

For this model, we use the Python module SciPy's least-squares optimisation with the Trust Region Reflective algorithm. It gives an optimised value for inverted parameters and a covariance matrix from which we derive an uncertainty for each parameter. To prevent $p < -1$ and $H_{\mathrm{m}} < 0$, we implement the scheme using $p = e^{p'} - 1$, also done in Parrenin et al. (2017), and $H_{\mathrm{m}} = e^{H'_{\mathrm{m}}}$. The three inverted parameters are the surface accumulation rate $\bar{a}$, the Lliboutry profile parameter $p'$ and the log of the mechanical ice thickness $H'_{\mathrm{m}}$. The minimised cost function uses the least-squares expression:

$$S = \sum \frac{\left(\chi_i^{\mathrm{iso}} - \chi^{\mathrm{mod}}\left(d_i^{\mathrm{iso}}\right)\right)^2}{\left(\sigma_i^{\mathrm{iso}}\right)^2} + \frac{\left(p'_{\mathrm{prior}} - p'\right)^2}{\left(\sigma^{p'}\right)^2}, \tag{4}$$

where $\sigma^{p'} = 1$, allowing the function to vary within reasonable limits, and $p_{\mathrm{prior}} = 3$, a similar value to EDC. The depths and ages of isochrones are $d_i^{\mathrm{iso}}$ and $\chi_i^{\mathrm{iso}}$, respectively, $\sigma_i^{\mathrm{iso}}$ is the age uncertainty, and $\chi^{\mathrm{mod}}$ is the modelled age.

Isochrones, which have been traced and dated from radar observations, constrain the optimisation in the model (see Sect. 3). In order to assess the suitability of the model in representing the observed isochrones at each spatial position, we calculate the standard deviation of the residuals between observed isochrone ages and modelled ages as

$$\sigma_{\mathrm{R}} = \sqrt{\frac{\boldsymbol{R}^T\boldsymbol{R}}{n^{\mathrm{iso}}}}, \quad \boldsymbol{R} = \frac{\left(\overline{\chi}^{\mathrm{iso}}\right) - \left(\overline{\chi}^{\mathrm{mod}}\right)}{\overline{\sigma}^{\mathrm{iso}}}, \tag{5}$$

where $\boldsymbol{R}$ is a vector of the residuals, and $n^{\mathrm{iso}}$ is the number of isochrones. If the model is a good fit, then the "reliability index" $\sigma_{\mathrm{R}}$ is close to 0. However, if $\sigma_{\mathrm{R}} > 2$, it means the model is not an accurate representation of the observations. This often occurs in areas such as the flank of the dome or over trenches in the bedrock where horizontal ice flow is an important factor. Further details on this model can be found in the Supplement.

## 2.4 Bayesian information criterion (BIC)

The BIC is used to determine the relevance of one model relative to another (Kass and Rafterty, 1995; Legrain et al.,

2023). Specifically, we assess whether the inclusion of a stagnant ice layer in the model produces a better fit to the isochrone constraints than a model which does not allow for a stagnant ice layer. The BIC $C_{\text{bay}}$ is calculated by

$$C_{\text{bay}} = -2\ln L + K\ln N_{\text{iso}}, \tag{6}$$

where $N_{\text{iso}}$ is the number of isochrones used to constrain the model, and $L$ is the maximum log-likelihood, defined here as $\sigma_R N_{\text{iso}}$. The comparison of the relevance of model $i$ over model $j$ is then calculated as

$$\Delta C_{ij} = C_i - C_j. \tag{7}$$

The dominance of model $i$ over model $j$ is weak if $\Delta C_{ij} < 2$, positive if $2 < \Delta C_{ij} < 6$, strong if $6 < \Delta C_{ij} < 10$ and very strong if $\Delta C_{ij} > 10$.

## 3 Datasets

In this study we make use of three radar surveys collected in the LDC region, which were taken during the Antarctic field seasons from 2016–2020 and informed the selection of the location of the Beyond EPICA drill site (BELDC). As three different radar systems were used, we treat each dataset independently and compare the model results obtained. We focus on three different scales which correspond to the differing spatial coverages of the three surveys over LDC. We also look at the NP site which was investigated in one of the surveys and was initially tagged as a site of interest for Oldest Ice (Parrenin et al., 2017). In Fig. 2, we mark BELDC (75.29917° S, 122.44516° E) and the site selected for the Million Year Ice Core project drilling (MYIC; 75.34132° S, 122.52059° E).

Internal reflecting horizons (IRHs) were traced in each of the datasets in the two-way travel time domain. The depths of these IRHs were then calculated using the electromagnetic wave velocity in ice of 168.5 m µs$^{-1}$ as in Winter et al. (2017) and a firn correction of 10 m with an uncertainty of ±3 m (Lilien et al., 2021). Table 1 summarises the key characteristics of each dataset. In this section, we also briefly describe an autonomous phase-sensitive radio-echo sounder (ApRES) used to determine the vertical velocity of the ice at two selected locations, at EDC and near BELDC. These measurements give us an insight into the internal ice deformation, offering further evidence of a potential stagnant ice layer.

### 3.1 UTIG HiCARS

The Oldest Ice candidate A (OIA) survey was conducted in January 2016 by the University of Texas at Austin Institute for Geophysics (UTIG), the Australian Antarctic Division (AAD) and the French Polar Institute Paul-Émile Victor (IPEV) as part of the ICECAP project (International Collaborative Exploration of the Cryosphere through Airborne Profiling, Cavitte et al., 2016). Radar data were collected with the High Capability Airborne Radar Sounder (HiCARS) 1 and 2, operating over frequency range 52.5–67.5 MHz (Cavitte et al., 2021), and were processed and published in Young et al. (2017) and Cavitte et al. (2020, 2021). These data cover an area around $100 \times 100$ km$^2$ over LDC and Dome C. Cavitte et al. (2021) traced 26 IRHs, and 19 of those were dated by linking them to the EDC site using a different transect which passes closer to EDC than the HiCARS transect (Table 3 of Cavitte et al., 2021, provides IRH ages and uncertainties). The IRH depths near EDC range from $308.2 \pm 3.2$ to $2644 \pm 12$ m and ages range from $10.0 \pm 0.3$ to $366.5 \pm 7.9$ ka. In this study, we use the 19 dated IRHs, which we re-interpolate spatially to a point every 100 m for input into the 1D model, in order to obtain simulation results at a reasonable resolution while optimising computation time.

### 3.2 BAS DELORES

Over the Antarctic field seasons 2016–2017 and 2017–2018, the sledge-borne Deep Looking Radio Echo Sounder (DELORES) from the British Antarctic Survey (BAS) was used to explore two potential drill sites around the Dome C area, identified from the HiCARS survey results. The LDC area is covered by a dense network of 120 radar transects, mostly organised as a grid, with denser spots in areas of particular interest. DELORES was the only radar system to survey NP at high spatial resolution, collecting a grid of 21 radar transects, which cover an area of approximately $5 \times 5$ km$^2$. We discuss the LDC and NP areas separately, but it should be noted that the same 20 IRHs were traced manually over all DELORES radar transects using the Petrel E&P software. For the model, we re-interpolate the horizons every 10 m.

In order to date the IRHs, LDC is linked to the EDC ice core site through three independent radar transects, while NP is linked through one transect to EDC. Ages for the IRHs were then calculated by linearly interpolating the EDC age–depth timescale from AICC2012 (Bazin et al., 2013), where the IRHs are closest to the ice core site. As the depth difference between values in the AICC2012 dataset is 1.65 m, there is no need for more complex interpolation, as the uncertainty is negligible relative to other factors. The age of each IRH is obtained by first calculating the average of the ages obtained from the three EDC–LDC radar intersections and then averaging with the EDC–NP intersection. It is this average age for each IRH that is input to the model.

The depth uncertainty of each IRH arises from several contributions. The DELORES radar system operating over frequency range 0.6–7 MHz has a vertical resolution of 11.1 m and a range precision of 3 m (Cavitte et al., 2021). The two-way travel time to depth conversion results in a firn correction uncertainty of ±3 m (Lilien et al., 2021) and a ±0.44 % uncertainty for the wave speed in ice (Winter et al., 2017). We include an uncertainty of 100 ns, which corresponds to a depth uncertainty of ±8 m, due to error when tracing the IRHs. The final source of depth uncertainty is due to the slope of the isochrones and the distance between the EDC site and

**Table 1.** Summary table of radar datasets and traced IRH characteristics.

| Dataset | HiCARS[a] | DELORES | LDC-VHF |
|---|---|---|---|
| Location | Dome C and LDC | LDC and NP | LDC – Patches A and B |
| Coverage ($km^2$) | $100 \times 100$ | $20 \times 20$ (LDC) and $5 \times 5$ (NP) | $5 \times 8$ |
| Total length of survey transects (km) | 2825 | 1089 (LDC) and 152 (NP) | 158 |
| Closest distance to EDC (m) | –[b] | 338–404 | 178 |
| Transect spatial re-interpolation (m) | 100 | 10 | 10 |
| Number of IRHs | 19 | 20 | 19 |
| IRH depths closest to EDC (m)[c] | 308–2644 | 317–2740 | 1079–2826 |
| Overall IRH depth uncertainty (m)[c] | 3–12 | 11–23 | 12–18 |
| IRH ages (ka)[c] | 10.0–366.5 | 10.5–414.6 | 73.7–476.4 |
| IRH age uncertainties (ka)[c] | 0.3–7.9 | 0.5–10.5 | 2.2–14.7 |

[a] Cavitte et al. (2021). [b] Transects were not linked to EDC using a HiCARS radar line (Cavitte et al., 2021). [c] Range indicates values for shallowest to deepest IRH.

**Table 2.** IRHs traced in the DELORES radar dataset, average depth at the closest point to EDC, calculated average age and uncertainty.

| DELORES IRHs | Depth nearest EDC (m) | Age (ka) | Age uncertainty (ka) |
|---|---|---|---|
| | $z$ | $\chi$ | $\sigma_\chi$ |
| IRH_1 | 317 | 10.5 | 0.5 |
| IRH_2 | 606 | 30.4 | 1.5 |
| IRH_3 | 705 | 39.2 | 1.1 |
| IRH_4 | 798 | 47.1 | 1.4 |
| IRH_5 | 961 | 62.2 | 2.4 |
| IRH_6 | 1084 | 75.2 | 2.2 |
| IRH_7 | 1136 | 80.0 | 1.9 |
| IRH_8 | 1169 | 82.6 | 1.9 |
| IRH_9 | 1339 | 97.7 | 2.1 |
| IRH_10 | 1454 | 108.3 | 2.5 |
| IRH_11 | 1682 | 128.5 | 2.0 |
| IRH_12 | 1903 | 166.1 | 4.7 |
| IRH_13 | 2077 | 202.3 | 3.5 |
| IRH_14 | 2263 | 240.4 | 3.8 |
| IRH_15 | 2344 | 262.2 | 6.9 |
| IRH_16 | 2526 | 326.2 | 6.6 |
| IRH_17 | 2592 | 345.5 | 10.1 |
| IRH_18 | 2659 | 381.5 | 12.3 |
| IRH_19 | 2687 | 395.8 | 10.0 |
| IRH_20 | 2740 | 414.6 | 10.5 |
| Bedrock | 3198 | – | – |

the point of closest approach of each of the four radar transects used for dating (338–404 m). The gradient of each IRH over the last traceable 100 m is extrapolated over this gap, resulting in an uncertainty contribution of 1 to 15 m, increasing with depth. Combining the various sources of uncertainties gives an overall depth uncertainty of 12 m for the shallowest IRH to 23 m for the deepest IRH. The calculated ages for each horizon are in Table 2, along with uncertainties which take into account that of the depth and of the AICC2012 age–depth profile (Bazin et al., 2013).

## 3.3 UA LDC-VHF

In the 2019–2020 Antarctic field season, a multichannel coherent depth sounder operating in the VHF frequency range was deployed to survey the potential Beyond EPICA drill sites in the LDC area, known as Patch A and Patch B (Fig. 2c; Yan et al., 2020; Lilien et al., 2021). Due to logistical and time constraints, a polarimetric ultra-wideband system operating over 170–470 MHz and a new ultra-high frequency system operating at 600–900 MHz with a large array in a Mills Cross array configuration could not be deployed. Those systems were developed by The University of Alabama (UA), the University of Copenhagen (CPH) and the Alfred Wegener Institute (AWI). UA also developed a high-power radar operating at 200 MHz with 60 MHz bandwidth to complement and supplement data from the other radars, and this system was used in 2019–2020. It was capable of transmitting 8 µs chirped pulses with 500 W peak power per channel at a pulse repetition frequency of 10 kHz (Yan et al., 2020). This system was operated with much reduced sensitivity because of the failure of the balloon carrying the antenna structure and the compact power generator housed close to the radar system. The generator failure led to a large secondary generator being mounted on a metal structure between the vehicle and the antenna arrays. A long power cable powered the system in the tracked vehicle, and radio-frequency cables were forced to route next to the generator. These complications forced the field team to reduce the peak power to less than 100 W for each channel and operate only four channels to reduce radio-frequency interference (RFI) issues. In the following, we will refer to data collected using this system as the LDC-VHF radar dataset. The dataset consists of 12 transects covering Patches A and B with parallel lines. In this study, we used the seismic environment of the Section software from Paradigm Geophysical to trace 19 IRHs along the 12 radar transects and re-interpolated them to a 10 m horizontal spacing. A delay time of 1 µs was required during processing and was confirmed by matching the reflectivity pattern

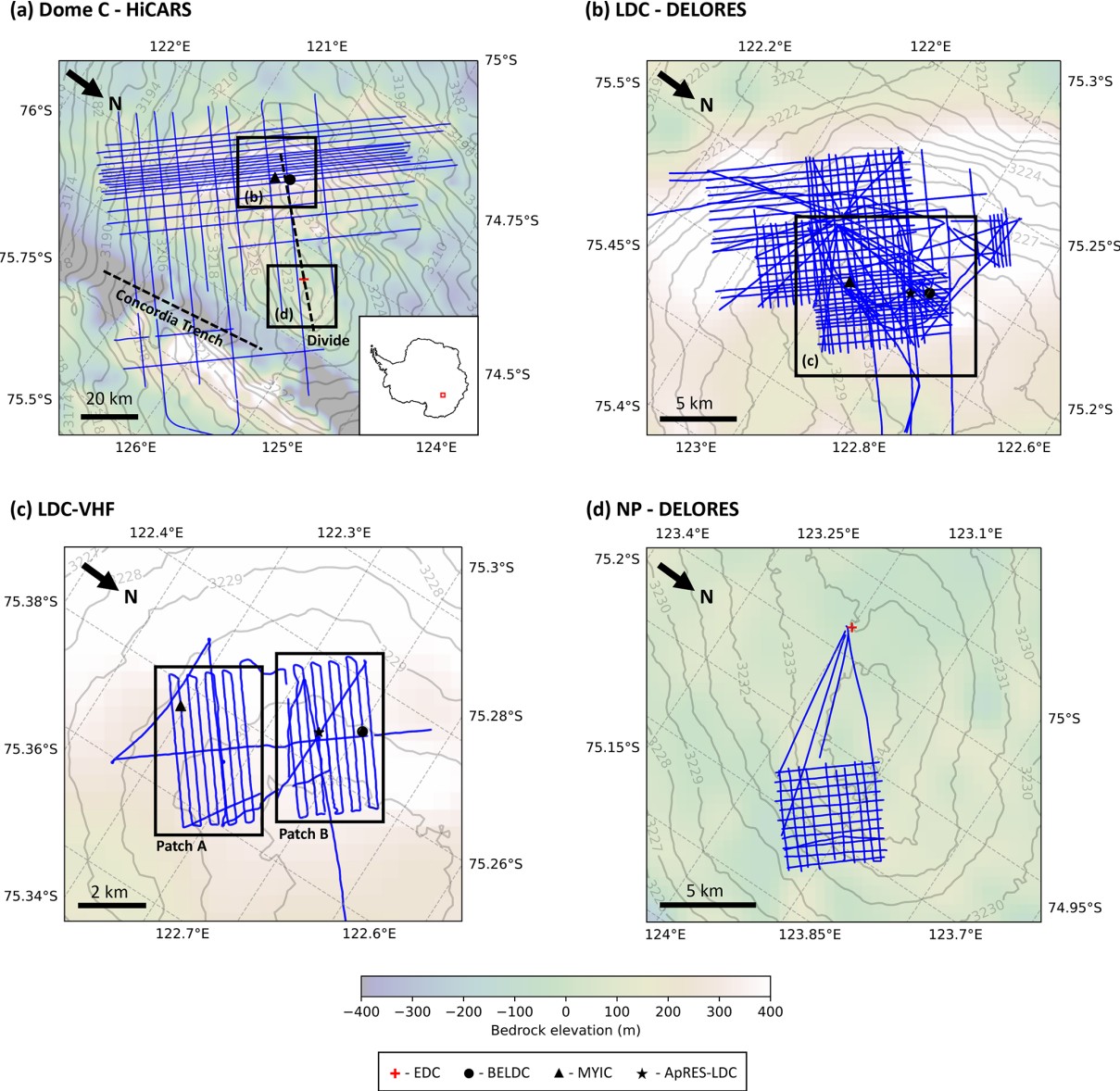

**Figure 2.** Maps showing the four areas of interest near Dome C, with bedrock elevation from BedMachine version 3 (Morlighem, 2022) and surface elevation from REMA (Howat et al., 2019). Radar transects of each survey are in blue. Ice core drill sites EDC, BELDC, MYIC and the ApRES-LDC site are marked by a red cross, black circle, black triangle and black star, respectively. **(a)** The larger Dome C region covered by the HiCARS airborne radar transects. The dashed black lines show the Concordia trench and Dome C divide. In the inset, the red square shows the location of Dome C in Antarctica. **(b)** LDC DELORES ground-based radar dataset, **(c)** LDC-VHF high resolution ground-based radar dataset and **(d)** NP DELORES ground-based dataset.

and bedrock reflection at the EDC end of the profile to those of the radar data presented in Winter et al. (2017). The IRHs were then dated using the direct radar transect link from LDC to the EDC ice core site. These IRHs and bedrock reflections were traced and dated independently of Lilien et al. (2021), who focused only on the EDC–LDC transect.

There are two components which contribute to the isochrone age uncertainty. The first comes from the dating of the EDC ice core in AICC2012 (Bazin et al., 2013, Table 2).

The second component, the depth uncertainty, includes four factors: 3 m firn depth uncertainty, 0.44 % uncertainty in the wave speed in ice, 11 m human error in IRH tracing and the gradient uncertainty over the 178 m gap from the transect to EDC (1 to 6 m). The overall depth uncertainty ranges from 12 to 18 m from the shallowest to the deepest IRH. The calculated ages and uncertainties are shown in Table 3.

The high vertical resolution of the radar data shows a layer of ice just above the bedrock where there is an abrupt change

**Table 3.** IRHs traced in LDC-VHF dataset, depths at the closest point to EDC, calculated ages and uncertainties.

| LDC-VHF IRHs | Depth nearest EDC (m) | Age (ka) | Age uncertainty (ka) |
|---|---|---|---|
| | $z$ | $\chi$ | $\sigma_\chi$ |
| IRH_1 | 1079 | 73.7 | 2.2 |
| IRH_2 | 1206 | 84.6 | 1.9 |
| IRH_3 | 1270 | 90.3 | 1.9 |
| IRH_4 | 1342 | 96.9 | 2.0 |
| IRH_5 | 1507 | 113.5 | 2.1 |
| IRH_6 | 1596 | 121.3 | 1.9 |
| IRH_7 | 1747 | 132.9 | 2.5 |
| IRH_8 | 1889 | 160.5 | 4.3 |
| IRH_9 | 1977 | 180.1 | 3.6 |
| IRH_10 | 2095 | 203.1 | 3.0 |
| IRH_11 | 2165 | 215.2 | 3.1 |
| IRH_12 | 2274 | 240.2 | 3.3 |
| IRH_13 | 2296 | 243.7 | 2.9 |
| IRH_14 | 2486 | 304.9 | 5.9 |
| IRH_15 | 2525 | 321.2 | 5.6 |
| IRH_16 | 2584 | 337.0 | 4.8 |
| IRH_17 | 2646 | 367.4 | 9.4 |
| IRH_18 | 2706 | 397.8 | 7.5 |
| IRH_19 | 2826 | 476.4 | 14.7 |
| Bedrock | 3239 | – | – |

in the return power and lateral coherency of the radar signal. In this layer, there are no visible continuous IRHs, although the radar system is effective at these depths, as the signal-to-noise ratio is sufficiently high to detect continuous IRHs. This layer was first presented in Lilien et al. (2021) and termed the basal unit. It is visible in other datasets with varying clarity (e.g. Winter et al., 2017) but in particular also in the HiCARS data (Cavitte, 2017). In this study, we traced the depth of the top of the basal unit where it was visible in the radargrams.

## 3.4 BAS/UCL ApRES

During the 2016–2017 and 2017–2018 field seasons, the autonomous phase-sensitive radio-echo sounder (ApRES) was also used in two locations to explore englacial flow in a collaboration between BAS and University College London (UCL). Nicholls et al. (2015) give an overview of the system, data processing workflow and applications that we simply summarise here for convenience.

ApRES is a ground-based radar system capable of tracking temporal displacement in the position of IRHs. The position changes are transformed from travel time to depth following Kingslake et al. (2016) using the density profile measured at EDC and assuming it applies at LDC (Le Meur et al., 2018). Due to the particularities of basal ice in the area, ApRES was capable of detecting reflectivity events as close as 20 m from the bed in the two successive field seasons (Fig. 3). These

events were used as a reference to transform IRH vertical displacement into full-depth vertical velocity. ApRES measurements were taken at EDC and at a location at LDC, Patch B near BELDC (ApRES-LDC, star in Fig. 2, 75.30832° S, 122.469022° E).

## 4 Results

### 4.1 ApRES measurements

In order to study the vertical stress on the ice, we first look at ApRES measurements in the Dome C area. The ApRES-derived full-depth vertical velocity at the ApRES-LDC site is shown in Fig. 3a. For reference, we show the englacial vertical velocity at EDC (Fig. 3b) derived previously also using ApRES and presented in Buizert et al. (2021). For the two sites, the best-fit Lliboutry velocity profile (Eq. 3) was determined using the ordinary least-squares method.

Figure 3 shows near-surface compaction and almost uniform vertical strain rate until almost 2000 m depth for ApRES-LDC and a few hundred metres deeper for EDC. Below these depths, we are unable to track reflections. This is expected, as the radar signal coherence generally deteriorates with depth. However, near the bed, at around 160 m above the bed for ApRES-LDC and about 20 m for EDC, we manage to track a few strong reflections. We follow the standard ApRES uncertainty estimation technique described in Brennan et al. (2014) and Nicholls et al. (2015), where phase uncertainty depends on signal-to-noise ratio. The uncertainty inversely relates to signal-to-noise ratio and is lower in the basal unit than in the ice above it, which is unusual as it typically decreases with depth. The presence of stagnant ice is revealed by a comparison of ApRES data from the two locations. For the ApRES-LDC site, the magnitude of the vertical velocity decreases rapidly and reaches close to zero at a depth of 2000 m. In the top 2000 m of the ice column, we have a high density of ApRES data points, and our uncertainty is smaller than at the EDC site. If we assume that the absolute value of vertical velocity is decreasing monotonically with depth, it follows that vertical velocity in the bottom section of the column must be smaller than at the EDC, and ice towards the base must be nearly or totally stagnant.

### 4.2 Distribution of the basal unit

Here, we investigate the extent of the potentially stagnant basal unit at LDC which was observed in the radar surveys. Features comparable to what we now refer to as the basal unit are visible in many radar datasets over Antarctica (often referred to as echo-free zone, e.g. Drews et al., 2009). However, in our region of interest, discussions so far only include a description from the HiCARS airborne radar data (Cavitte, 2017) and observations from a single radar line in the LDC-VHF radar survey (Lilien et al., 2021). Figure 4a shows the

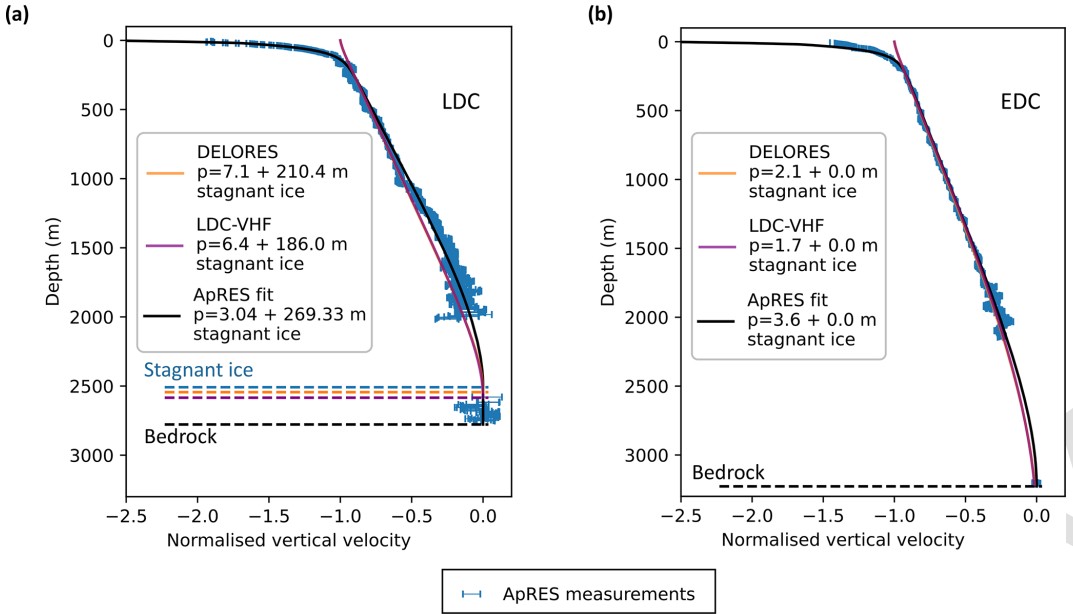

**Figure 3.** Full-depth vertical velocity (in blue) derived with ApRES in the proximity of **(a)** ApRES-LDC at Patch B near BELDC and **(b)** EDC. The best fit of the Lliboutry (1979) velocity profile (Eq. 3) to the ApRES data is shown in black. The parameters for the ApRES best fits are **(a)** for ApRES-LDC $p = 3.04$ with 269.33 m of basal stagnant ice, shown as a dashed blue line, and **(b)** for EDC $p = 3.6$ with no stagnant ice. Orange and purple lines correspond to the vertical velocity fits produced by the 1D model applied to the DELORES and LDC-VHF radar datasets, respectively. As the model results are similar for both datasets, the lines are almost superimposed.

depth of the top of the basal unit for the whole LDC-VHF dataset which was traced in this study.

Generally, the top of the basal unit is deeper towards the northeast and EDC. This is in large part due to the dip in 5 the local bedrock topography and therefore an increase in ice sheet thickness. In these places, the thickness of the basal unit remains fairly constant, therefore the increase in overall ice sheet thickness results in the increase in depth of the top of the basal unit. There are a couple of anomalous points which 10 do not seem to follow this trend. These come from a mixture of an unclear radar signal and the ambiguity in tracing the basal unit.

### 4.3 Relevance of including a stagnant ice layer in model and comparison to observations

15 Observations from radar surveys show that there appears to be a basal unit at LDC of unknown properties and origin. ApRES measurements show that the internal deformation of the ice sheet in the LDC area, as described by its vertical velocity profile, is better explained when an extensive stagnant 20 basal layer is present. Our model determines the age distribution of the ice column, constrained by radar-observed IRHs. In this section, we assess the relevance of the 1D numerical model, which allows for a layer of stagnant ice vs. a model with the observed bedrock as the lower age constraint, as in Parrenin et al. (2017). We apply this analysis to the LDC- 25 VHF dataset, as we have mapped the basal unit extent of

this area (Sect. 4.2). Figure 5 shows the $\Delta C_{ij}$ values (see Sect. 2.4) for the LDC-VHF radar dataset. The average $\Delta C_{ij}$ value is 4.1, which indicates that the stagnant ice model is more relevant than the non-stagnant ice model. Therefore we 30 use the stagnant ice model throughout the rest of this article.

Figure 6 shows the reliability index ($\sigma_r$, Eq. 5) across all four areas of interest that we investigate in this study. The reliability index is less than 2 in most of the surveyed area and even less than 1 over LDC and NP, which indicates that 35 the model fits to within $1\sigma$ of the IRH age constraints.

In order to compare the output of the 1D stagnant ice model to observations, in Fig. 3 we also show the vertical velocity profile determined according to the 1D model for the closest points to the ApRES measurements in the DE- 40 LORES and LDC-VHF datasets. The vertical velocities include not only ice vertical compaction and compression but also surface thinning or thickening and the vertical component of the along-surface advection. As the 1D model and ApRES data integrate over different time spans, we com- 45 pare the shape functions that do not depend on those processes, which change with time, despite our pseudo-steady assumption. We normalise the vertical velocities to the value at 150 m, which is below the depth of firn compaction. We can see from Fig. 3 that the model fits the observations well 50 at both ApRES sites. One possible reason for the different $p$ values for the ApRES fit and 1D model could be the different constraints used. For the ApRES data at EDC, the deepest return excluding the basal reflection was at 2145 m,

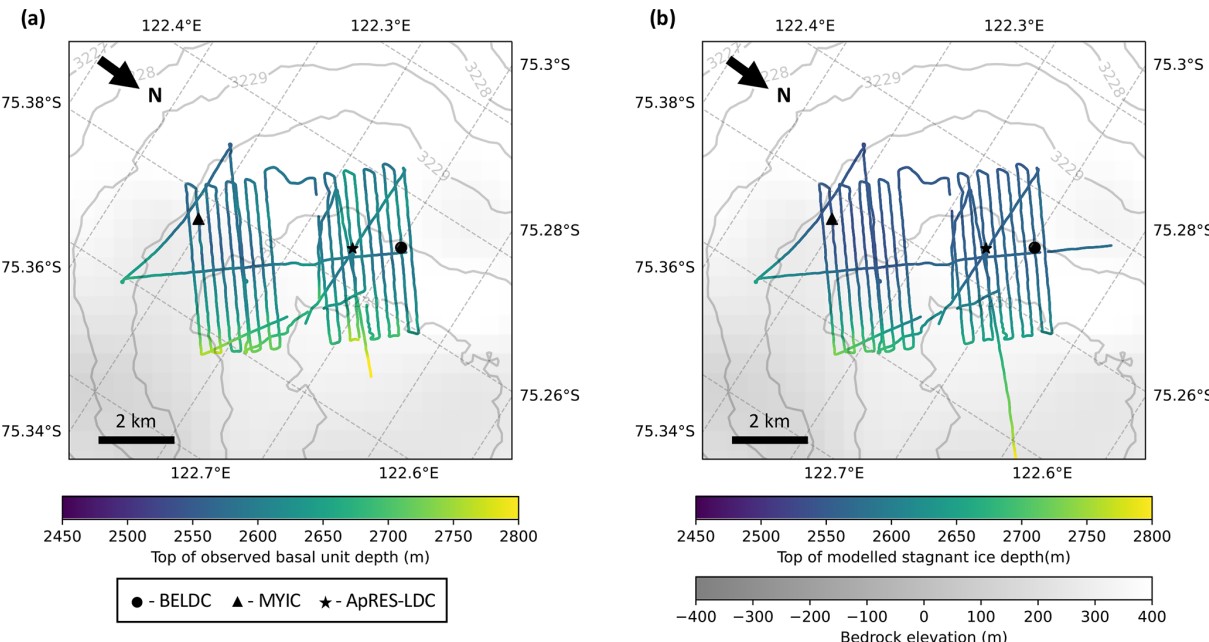

**Figure 4. (a)** Depth of the top of the basal unit traced in the LDC-VHF radar dataset, **(b)** depth of the top of the modelled stagnant ice.

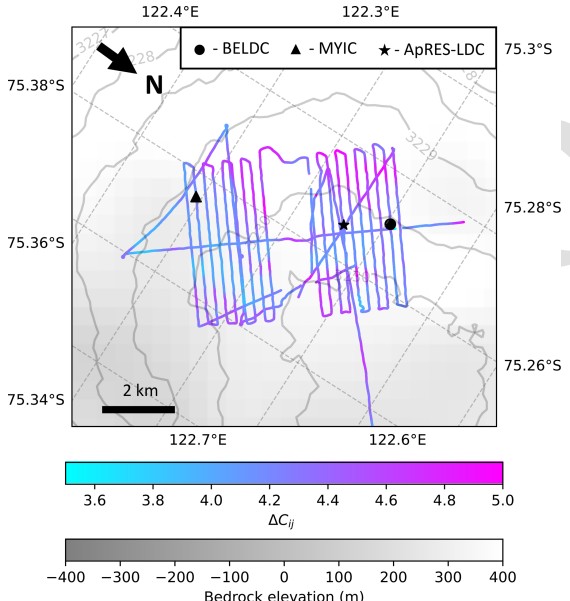

**Figure 5.** The $\Delta C_{ij}$ value which compares the relevance of the model, which inverts $H_m$ and the model, and which uses $H_{obs}$ as a constraint, applied to the LDC-VHF radar dataset.

whereas the deepest IRHs traced in the radar surveys was 2826 m. Having deeper constraints for the 1D model significantly affects the non-linear portion of the thinning function and therefore the $p$ value (Parrenin et al., 2017).

Figure 4b displays the depth of the top of the modelled stagnant ice layer for the LDC-VHF dataset. This depth is equal to the mechanical ice thickness $H_m$ (Fig. 1). It follows the same trend as the observed basal unit (Fig. 4a), with the deeper values occurring where the bedrock is also deeper. Comparing Fig. 4a and b, it is clear that the depths of the observed basal unit and the top of the modelled stagnant ice are similar. Figure 7a shows the modelled age of a single radar transect in the LDC-VHF dataset. From this, we can see that although the stagnant ice depth does not exactly match that of the observed basal unit, there is a remarkably close relationship between them, given that these values were modelled and observed independently.

Figure 7b shows the difference between the modelled stagnant ice and the observed basal unit depths. Throughout most of the radar survey, the stagnant ice thickness is larger by an average of 40 m, with a standard deviation of $\pm 20$ m. This corresponds to an average difference of approximately 1.5 % relative to the total ice thickness with a standard deviation of $< 1 \%$. The locations with larger differences correspond to the areas where the basal unit appeared to thin when tracing. We should note here that tracing the top of the basal unit in the LDC-VHF radar dataset is ambiguous due to the diffuse nature of the boundary between coherent and incoherent radar returns; different manual interpretations of where this top boundary is can differ by tens of metres.

### 4.4 Modelled stagnant ice and melting

Our model uses an inverted mechanical ice thickness $H_m$ (Fig. 2) to infer either a basal melt rate or a layer of stagnant ice. In Fig. 8, we show the results of the stagnant ice thickness or basal melt rate for the four areas of interest cov-

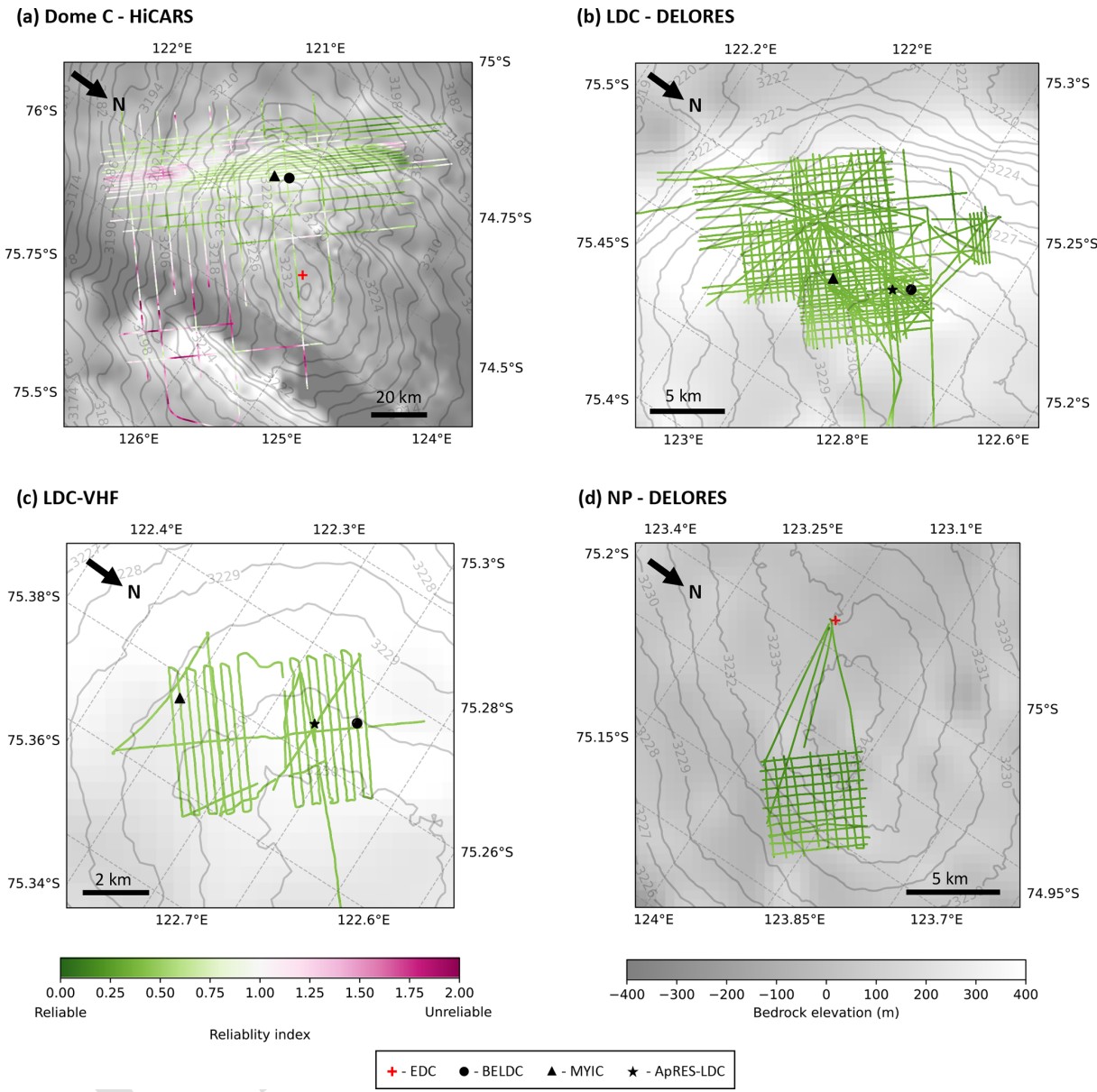

**Figure 6.** Reliability of the model, from 0 green (reliable) to 2 dark pink (unreliable); both colour bars correspond to all four panels. **(a)** HiCARS airborne dataset, **(b)** LDC DELORES ground-based radar, **(c)** LDC-VHF high resolution ground-based radar dataset and **(d)** NP DELORES ground-based dataset.

ered by the radar surveys (see Sect. 3). From the HiCARS dataset (Fig. 8a), we see that the model predicts a layer of stagnant ice on the LDC bedrock relief. There is significant melting predicted around the edges of the LDC relief, espe-
5 cially on the western side of LDC and across the Concordia trench (Fig. 2a), where Fig. 6a shows that the model is less reliable. These areas are more subject to horizontal flow, implying that the 1D assumption is not valid in that case. There is a low melt rate on the plateau around EDC which agrees
10 with the findings of the EPICA drill project (EPICA mem-

bers, 2004; Parrenin et al., 2007; Tison et al., 2015; Passalacqua et al., 2017, 2018).

In Fig. 8b and c, we zoom into the LDC area. The DELORES and LDC-VHF radar datasets confirm that there is stagnant ice in this region. Both datasets show stagnant ice 15 thicknesses up to ∼ 250 m. The DELORES dataset shows that the stagnant ice layer thins to near 0 m at the edge of the LDC relief, while the LDC-VHF dataset shows that near the Oldest Ice sites at Patch A and Patch B (MYIC and BELDC) the minimum thickness of the stagnant layer remains around 20 100 m.

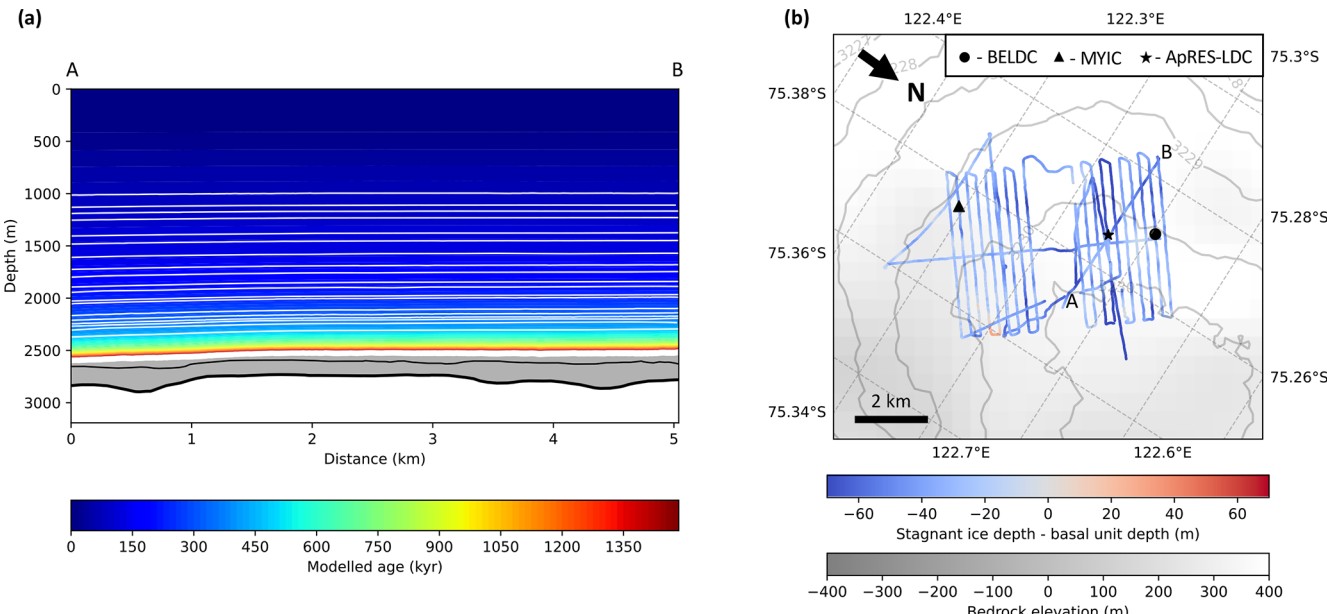

**Figure 7.** **(a)** A radar transect in the LDC-VHF dataset which passes diagonally across Patch B from A to B in panel **(b)**. The colour gradient shows modelled age, as shown in the side colour bar. The horizontal white lines represent the IRHs traced in the radar data, which constrain the model. The bold black line is the bedrock, the grey area is the modelled stagnant ice layer and the thin black line is the observed top of the basal unit from the radargram. All points along the radar transect are categorised as reliable according to Eq. (5). **(b)** Depth of modelled top of stagnant ice minus depth of top of the basal unit, as observed in the radar data.

**Table 4.** Modelled results for different radar transects at the points of closest approach to EDC. The observed bedrock depth for the DE-LORES NP transect is shallower than 3189 m; therefore, we give no age in the table.

|  | LDC DELORES | | | NP DELORES | LDC-VHF |
|---|---|---|---|---|---|
| Distance of closest point to EDC (m) | 368 | 338 | 338 | 404 | 178 |
| Total ice thickness (m) | 3239 | 3214 | 3200 | 3141 | 3239 |
| Age at 3189 m depth (ka) | $1015 \pm 87$ | $1013 \pm 86$ | $1016 \pm 86$ | – | $996 \pm 78$ |

In contrast to LDC, at NP, the mechanical thickness $H_m$ obtained from the model is remarkably similar to the observed bedrock depth from the radar $H_{obs}$, indicating a lack of melting or stagnant ice (Fig. 8d). There is an area of stagnant ice on the east side of NP, but its thickness is generally below 60 m. Where melting occurs, the values are below $0.1 \, \text{mm} \, \text{yr}^{-1}$. This low rate is not significant relative to its uncertainty which is between $0.05$–$0.1 \, \text{mm} \, \text{yr}^{-1}$.

### 4.5 Accumulation rate and $p$ parameter

In Fig. 9, we show the modelled temporally averaged ice-equivalent accumulation rate for each area considered. At LDC, we can see that the accumulation rate is generally around $19 \, \text{mm} \, \text{yr}^{-1}$. Looking at Fig. 9a, the HiCARS dataset shows that the accumulation rate is higher in surface depressions (dashed red circles) than in the surrounding areas. This is likely because snow is blown into the depressions from areas with higher surface elevation (Cavitte et al., 2018). The

accumulation at NP (Fig. 9d) is around $20 \, \text{mm} \, \text{yr}^{-1}$, slightly higher than at LDC.

Figure 10 shows how the Lliboutry profile $p$ parameter (Eq. 3) varies across the four surface areas surveyed by the radar systems. The $p$ parameter is quite high at LDC, with values between 5 and 8. Around the edges of LDC, Fig. 10a shows that $p > 8$. At NP (Fig. 10d), where there is little to no modelled stagnant ice, $p$ is much lower, between 2.2 and 3.6.

### 4.6 Model results at EDC

In order to evaluate the accuracy of the model as a predictor for ice core properties, we look at the modelled results for EDC. There were three radar transects in the LDC DE-LORES dataset, one transect in the NP DELORES data and one transect in the LDC-VHF radar dataset which pass close to the EDC drill site. Table 4 shows the model age result at the points of closest approach to EDC for each radar transect.

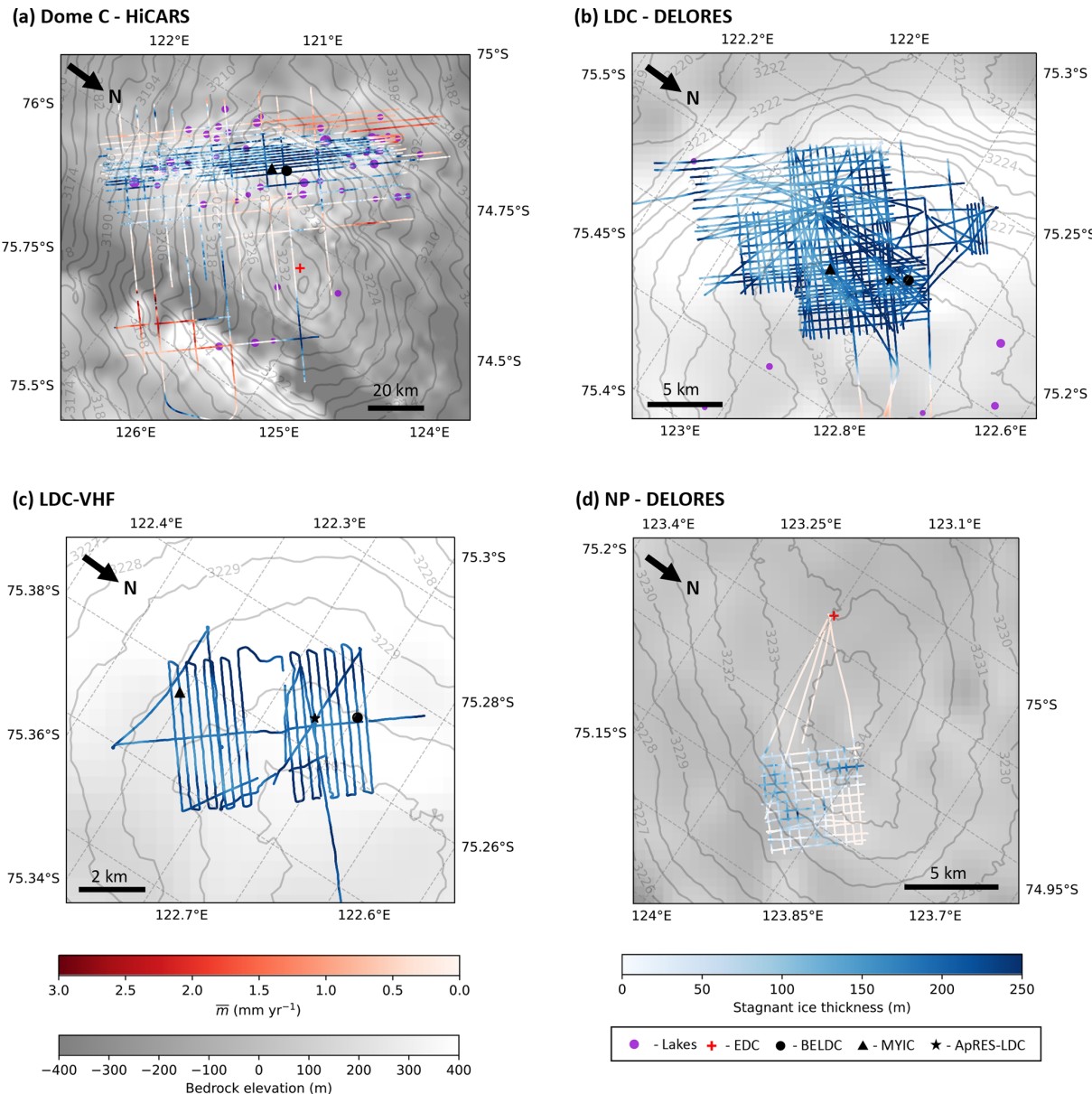

**Figure 8.** Modelled stagnant ice and basal melting results for each dataset, with the panels showing the same areas as in Fig. 2. Stagnant ice thickness, melt rate and bedrock elevation colour bars are the same for all four maps. Lakes are displayed as purple dots (Young et al., 2017; Livingstone et al., 2022). **(a)** HiCARS airborne dataset, **(b)** LDC DELORES ground-based dataset, **(c)** LDC-VHF high resolution ground-based dataset and **(d)** NP DELORES ground-based dataset.

The three LDC DELORES transects all start at approximately the same point, $\sim$ 338–368 m southeast of EDC, whereas the NP DELORES transect begins 404 m northeast of EDC. This difference can explain why the total ice thickness is 3141 m for the NP transect but ranges from 3200–3239 m for the LDC transects. The closest point to EDC in the LDC-VHF dataset is 178 m away, with a total ice thickness of 3239 m. The deepest ice which could be dated in the EDC ice core was at 3189 m depth, where the age inferred from the climatic signals is $801 \pm 9.6$ ka (Bazin et al.,

2013). At this depth, the modelled ages range from $1013 \pm 86$ to $1016 \pm 86$ ka for the DELORES dataset, and the age was found to be $996 \pm 78$ ka for the LDC-VHF dataset.

## 4.7 Oldest Ice site prospects around Dome C

Two deep ice core drilling campaigns are currently ongoing around the LDC area, so we focus on the age modelling results at the two sites. We define the maximum age as the age where the age density is sufficient for palaeoclimatic reconstructions with the measurement sensitivities currently

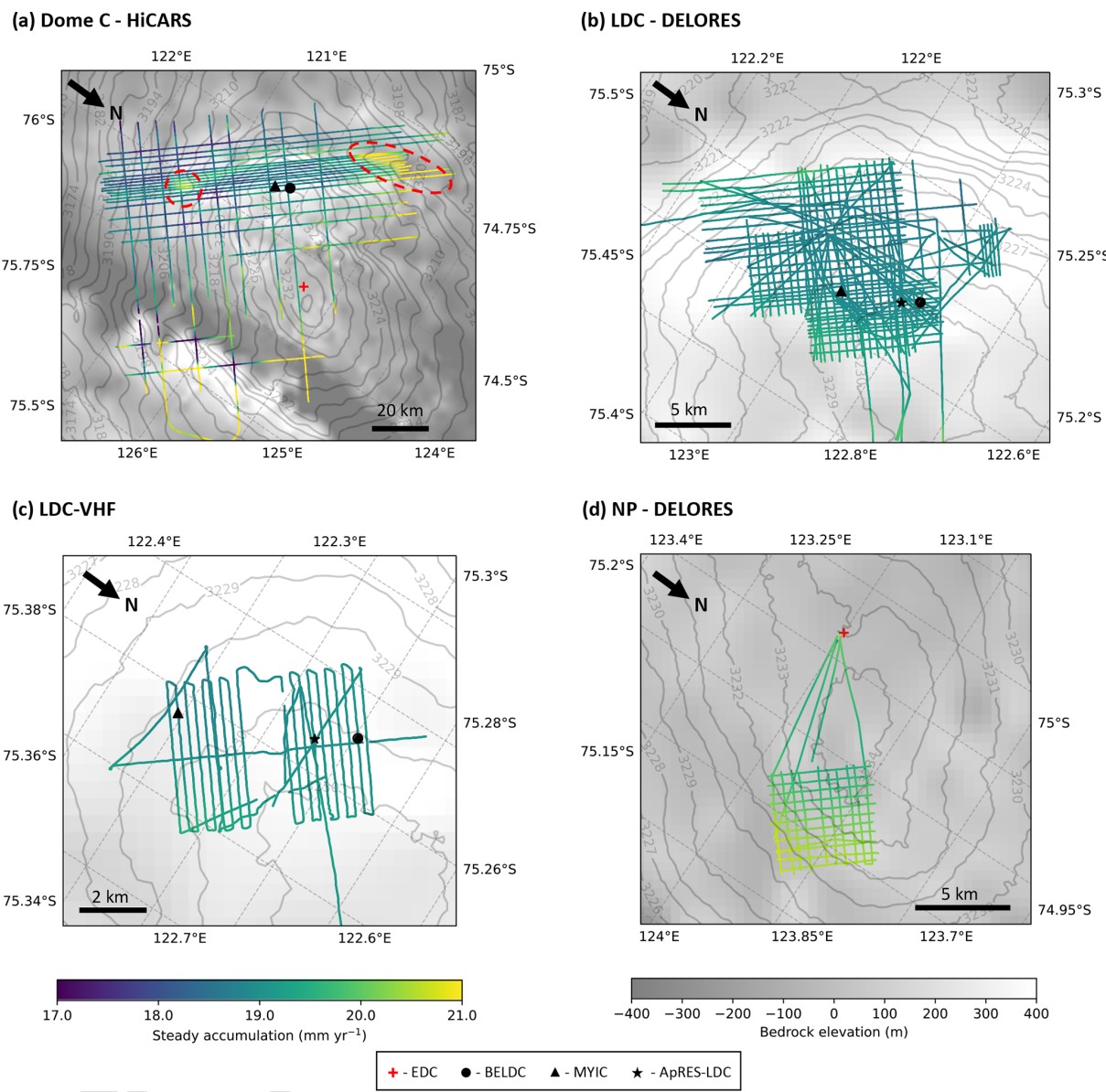

**Figure 9.** Modelled temporally averaged ice-equivalent surface accumulation rate for each study. Colour bars correspond to all four panels. **(a)** HiCARS airborne dataset, dashed red circles show local surface depressions which have higher accumulation rates than surrounding areas; **(b)** LDC DELORES ground-based dataset; **(c)** LDC-VHF high resolution ground-based dataset; and **(d)** NP DELORES ground-based dataset.

achievable, set at $20 \, \mathrm{kyr \, m^{-1}}$ (Fischer et al., 2013; Van Liefferinge and Pattyn, 2013). At locations where there is basal melting, the maximum age is the age at the depth of the observed bedrock $H_{\mathrm{obs}}$. Figure 11 shows the maximum age for each of the radar datasets around Dome C. Over the LDC bedrock relief, the maximum age is spatially homogeneous with values around 1.5 Ma (Fig. 11b). Around the southern and western edges of the LDC relief where melting was predicted (Fig. 8a), the age is much younger, $\sim 0.9$–1 Ma. The maximum age is also much younger across the divide from EDC to LDC and over the Concordia trench (Fig. 2a).

As reaching 1.5 Ma old ice is not always possible with a target condition of $20 \, \mathrm{kyr \, m^{-1}}$, we plot age density at 1.2 Ma. Age density is defined as the number of annual layers per metre in the ice column with units of $\mathrm{kyr \, m^{-1}}$. Figure 12a shows that the age density is generally low around the edges of Dome C, where there is a large amount of melting. Over the LDC relief, where there is stagnant ice, Fig. 12b and c show that the age density for 1.2 Ma ice is $11$–$14 \, \mathrm{kyr \, m^{-1}}$.

Figure 11d shows that the maximum modelled age over the whole radar grid which covers NP is around 2 Ma. There is a small area in the north corner of the grid where there is a local

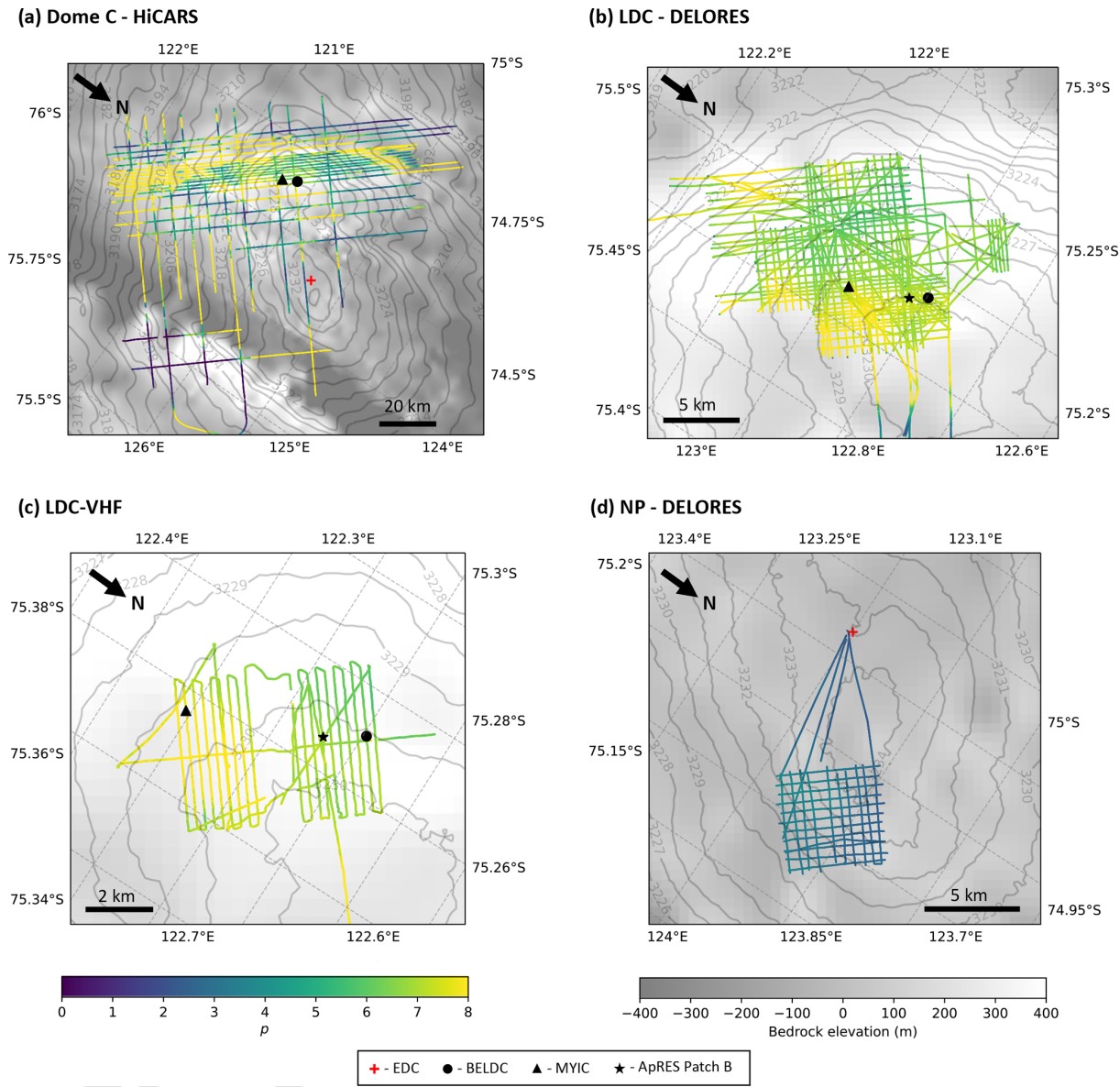

**Figure 10.** Llibotry profile $p$ parameter, colour bars correspond to all four panels. **(a)** HiCARS airborne dataset, **(b)** LDC DELORES ground-based dataset, **(c)** LDC-VHF high resolution ground-based dataset and **(d)** NP DELORES ground-based dataset.

bump in the bedrock; therefore, the bottom age is slightly younger than where it is smoother. Figure 12d shows that the age density of ice at 1.2 Ma is generally between 5 and 8 kyr m$^{-1}$ at NP. At 1.5 Ma, it is between 9 and 12 kyr m$^{-1}$.

## 4.8 Age and age density predictions at BELDC and MYIC

Both the DELORES and LDC-VHF radar transects are collected very close to the new BELDC drill site, at distances of 21 and 37 m, respectively. Therefore, using the model results, we can infer the potential properties of the core (see Table 5). The inverted mechanical ice thickness $H_{\mathrm{m}}$ is 2546 ± 30 and

2571 ± 27 m for the two datasets, which gives a thickness for the stagnant ice of 210 ± 35 and 186 ± 30 m, respectively. The top of the basal unit was traced 154 m above the bedrock in this study, and Lilien et al. (2021) gave a basal unit thickness of ∼ 200 m for the same radar line.

At BELDC, the maximum ages at 20 kyr m$^{-1}$ are 1.43 ± 0.16 Ma at 2484 m from the DELORES dataset and 1.47 ± 0.13 at 2504 m for the LDC-VHF dataset. Below the depth of this maximum age and above the stagnant ice, the remaining ice thickness is 62 and 67 m, respectively. Potentially, this older ice could still be useful for palaeoclimatic studies, if it is not too folded or disturbed. The age density of 1.2 Ma ice

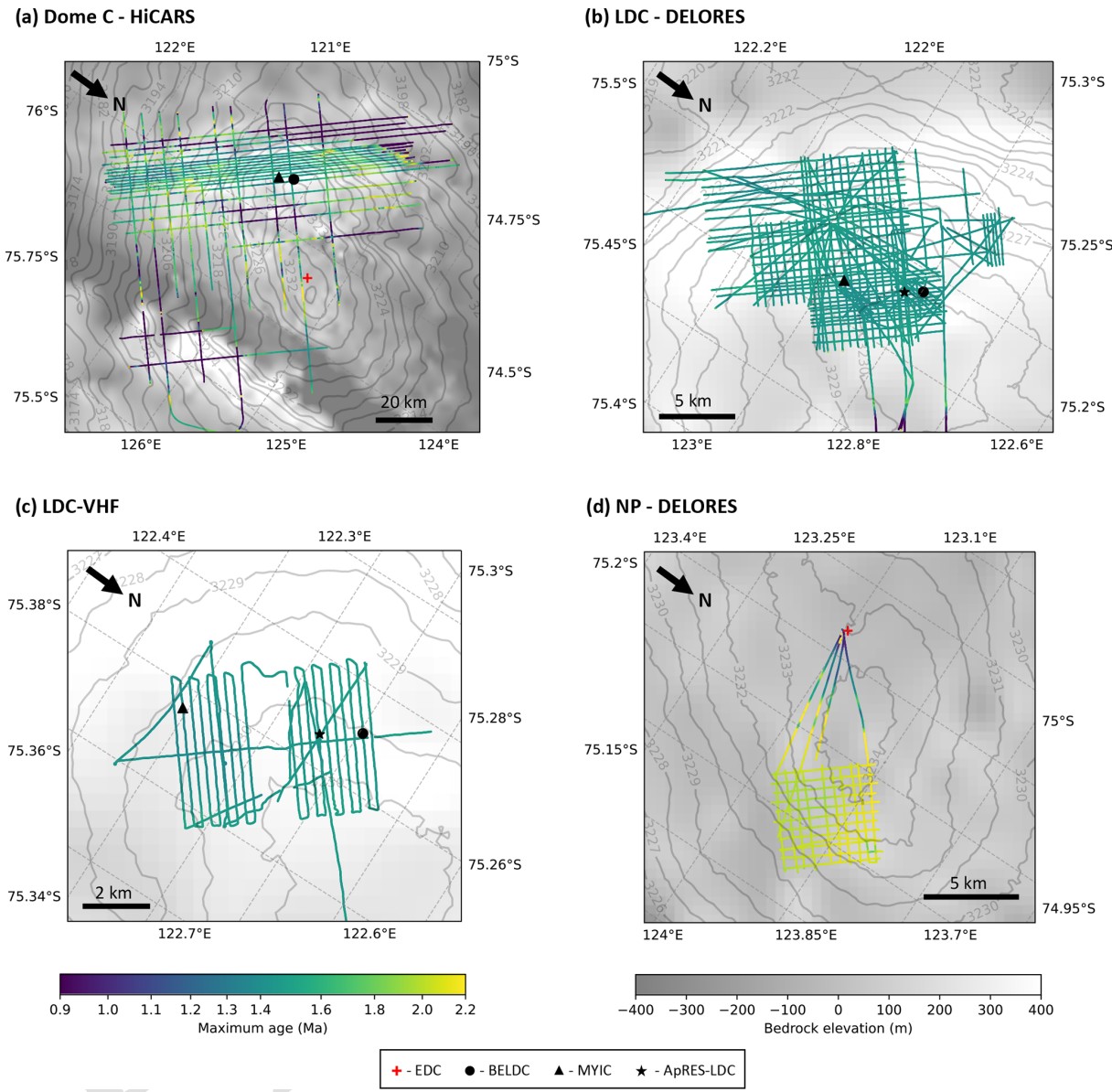

**Figure 11.** Maximum age with a maximum acceptable age density of $20\,\mathrm{kyr\,m^{-1}}$, colour bars correspond to all four panels. **(a)** HiCARS airborne dataset, **(b)** LDC DELORES ground-based dataset, **(c)** LDC-VHF high resolution ground-based dataset and **(d)** NP DELORES ground-based dataset.

is $13.2\,\mathrm{kyr\,m^{-1}}$ at a depth of 2469 m and $12.2\,\mathrm{kyr\,m^{-1}}$ at a depth of 2486 m, respectively.

Figure 13 shows the age profiles at BELDC from our two radar datasets, with the Lilien et al. (2021) results for com-5 parison. The age depth profiles of our results (DELORES, orange, and LDC-VHF, blue) are almost superimposed at depths shallower than 2200 m, showing that there is little variation in the predictions of ice age in this area. At depths below this (inset of Fig. 13), the profiles start to diverge. The 10 Lilien et al. (2021) (grey) and DELORES profiles (orange) are within each other's uncertainty ranges. The LDC-VHF profile (blue) predicts a slightly younger age at a given depth.

For comparison, we show the age–depth profile determined using the model which does not allow for a stagnant ice layer (black line). The profile obtained from modelling with no 15 stagnant layer clearly deviates from the IRH constraints (blue circles) at depths > 2200 m, supporting the conclusion that the inclusion of a stagnant ice layer in the model is more appropriate at LDC (Sect. 4.3).

Table 5 also shows the model results for the MYIC drill 20 site. With the exception of total ice thickness $H_{\mathrm{obs}}$, this site shows more consistent values between the DELORES and LDC-VHF datasets than are calculated at BELDC. The average values for all characteristics highlighted are within un-

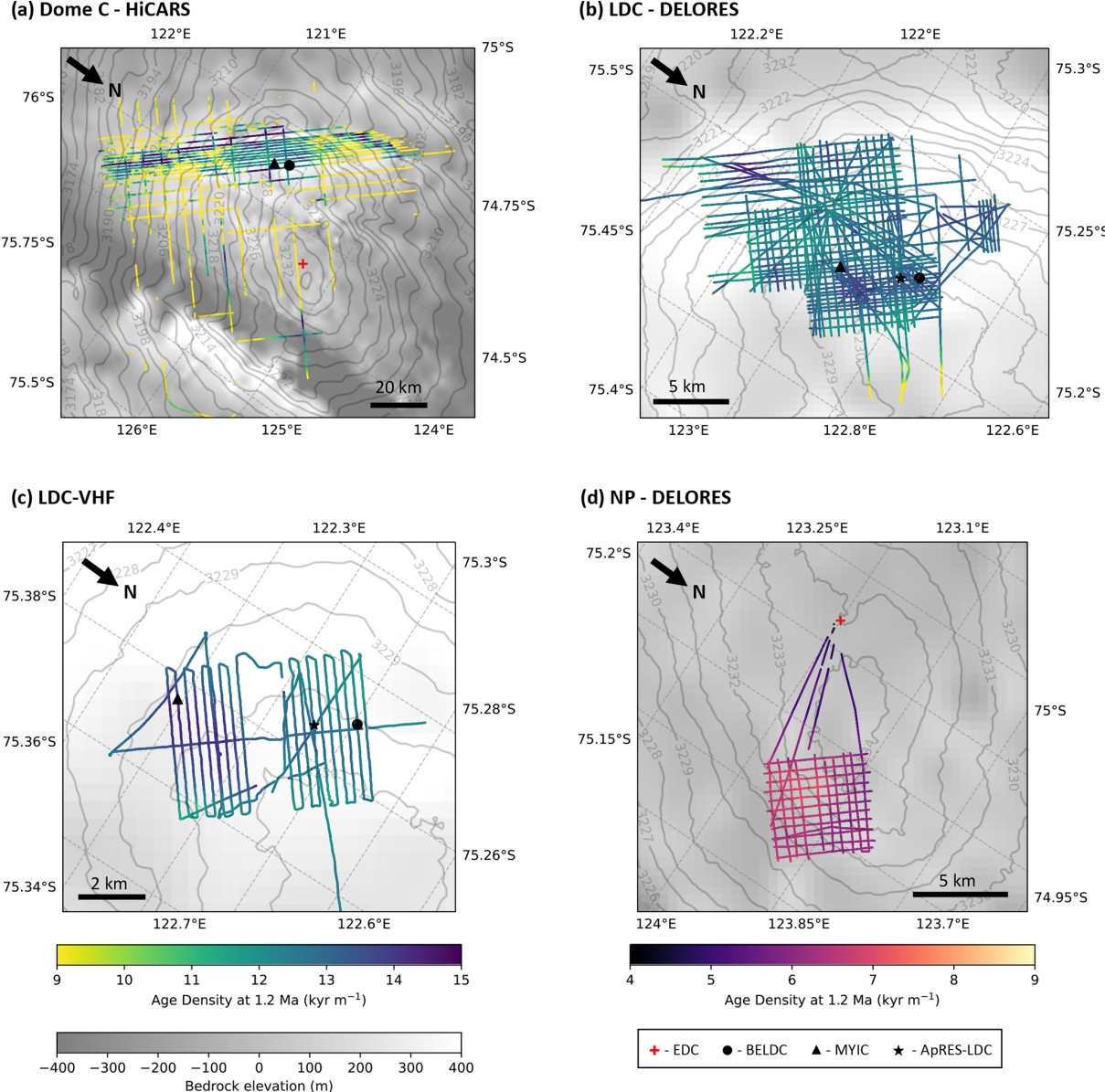

**Figure 12.** Age density at 1.2 Ma, the left hand colour bar corresponds to maps **(a–c)**, the right hand colour bar corresponds to panel **(d)** and the bedrock elevation colour bar corresponds to all four panels. **(a)** HiCARS airborne dataset, **(b)** LDC DELORES ground-based dataset, **(c)** LDC-VHF high resolution ground-based dataset and **(d)** NP DELORES ground-based dataset.

certainties for the two drill sites. An analogous age–depth profile figure for MYIC can be found in the Supplement (Fig. S1).

## 5 Discussion

### 5.1 Modelling limitations

In this section, we discuss the limitations when using the 1D numerical model. The novelty of our 1D model comes from the inversion of the bedrock depth from which we derive ei-

ther the basal melt rate or the thickness of a stagnant ice layer. In Sect. 4.3, we use the BIC to compare the relevance of this model with one which does not allow the bedrock depth to vary. The age–depth profiles at BELDC which are produced by these two models are shown in Fig. 13. These comparisons show that the model which allows the inclusion of a stagnant ice layer is more suitable for the Dome C region.

The pseudo-steady assumption of the 1D model means that the ratio between accumulation and melting is constant in time. The temporal accumulation variation $r(t)$ is calculated directly from the AICC2012 timescale (Bazin et al.,

**Table 5.** Modelling results for the BELDC and MYIC drill sites.

| | BELDC | | MYIC | |
|---|---|---|---|---|
| | DELORES | LDC-VHF | DELORES | LDC-VHF |
| Distance of closest point to drill site (m) | 21 | 37 | 55 | 0.5 |
| Total ice thickness $H_{obs}$ (m) | 2756 | 2757 | 2758 | 2742 |
| Mechanical ice thickness $H_m$ (m) | $2546 \pm 30$ | $2571 \pm 27$ | $2555 \pm 25$ | $2576 \pm 28$ |
| Stagnant ice thickness (m) | $210 \pm 35$ | $186 \pm 30$ | $202 \pm 35$ | $197 \pm 30$ |
| Depth of 1.2 Ma ice (m) | 2469 | 2486 | 2477 | 2472 |
| Age density at 1.2 Ma (kyr m$^{-1}$) | 13.2 | 12.2 | 13.0 | 13.8 |
| Depth of 1.5 Ma ice (m) | 2487 | 2506 | 2495 | 2489 |
| Maximum age at 20 kyr m$^{-1}$ (Ma) | $1.43 \pm 0.16$ | $1.47 \pm 0.13$ | $1.43 \pm 0.17$ | $1.39 \pm 0.14$ |
| Depth of max age ice (m) | 2484 | 2504 | 2492 | 2484 |
| Modelled $p$ value above stagnant ice | 7.1 | 6.4 | 7.2 | 7.9 |
| Temporally averaged accumulation rate (mm yr$^{-1}$) | 19.0 | 19.1 | 18.9 | 18.9 |

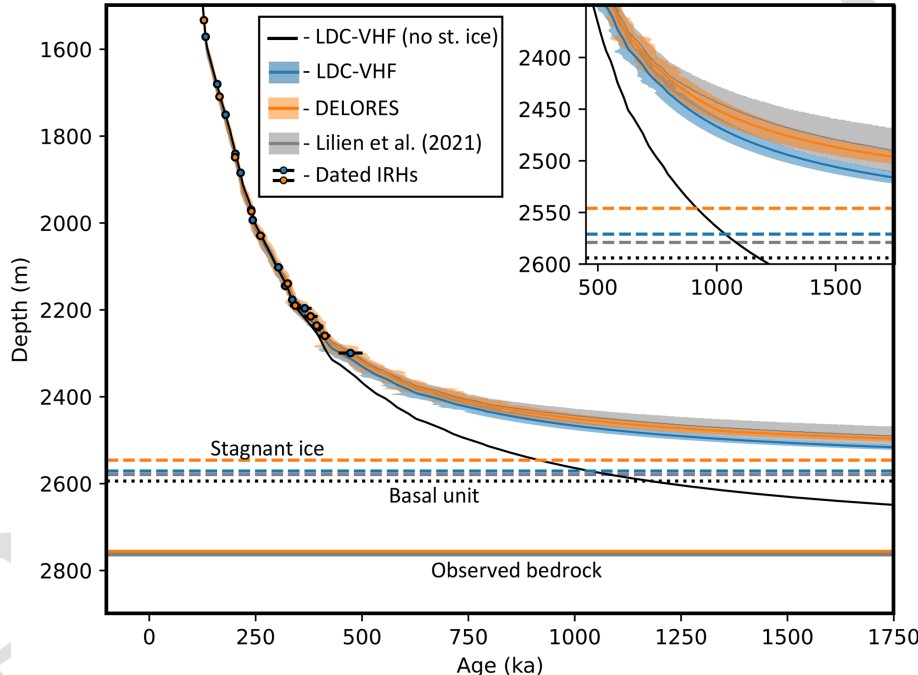

**Figure 13.** Modelled age–depth profile at the BELDC drill site along with age uncertainty. Results for the DELORES dataset are in orange, results for the LDC-VHF dataset are in blue, with the shaded areas showing $1\sigma$ uncertainty. The black circles are the IRH ages, with their age uncertainties shown as horizontal bars (Tables 2 and 3, respectively). The thick continuous lines show the radar-observed bedrock depths $H_{obs}$ (colours indicating the dataset of origin), the dashed lines provide the top of the modelled stagnant ice layer $H_m$ and the dotted black line shows the top of the basal unit identified in the LDC-VHF radar survey. The inset shows the deepest modelled section in more detail. Grey shows the result and $1\sigma$ uncertainty from modelling in Lilien et al. (2021) using IRHs continuously traced from BELDC to EDC. The stagnant ice depths for the LDC-VHF dataset and Lilien et al. (2021) are 2582 [TS1] and 2579 m, respectively, which is shown more visibly in the inset. The black line shows the modelled age–depth profile predicted by the model with no stagnant ice layer (Parrenin et al., 2017).

2013), so it is assumed to be identical to that inferred from the EDC ice core for all radar datasets. Of the available ice core records, EDC provides the most suitable accumulation record for the LDC and NP sites, as its proximity implies that it shares similar local conditions. Moreover, unlike the Vostok ice core, for example, where there is significant horizontal flow so deeper ice originates from further upstream than the shallower ice, EDC is relatively unaffected by local factors. EDC is $\sim 35$ km from LDC and $\sim 10$ km from NP, so the accumulation rates are not drastically different (Fig. 9; Le Meur et al., 2018). The oldest ice found at EDC was 800 ka; therefore, information on the accumulation variation $r(t)$ is limited to ice younger than this. Inferring accumulation variations from marine sediment cores (Lisiecki and

Raymo, 2005) for ice between 1500–800 ka, we find that the average accumulation rate is around 6 % higher during this time period than < 800 ka. This would correspond to an underestimation of layer thickness, which equates to ∼ 40 kyr overestimation in the modelled age of ice between 1500–800 ka. This overestimation is well within our uncertainty ranges for ice of this age.

The model does not account for any ice thickness variation over time, though this might only affect the thinning function and therefore inferred age by a few percent (Parrenin et al., 2007). The truncation of the vertical velocity profile implies that basal sliding can be present when there is basal melting, but it is not calculated independently. Figure 12 shows that the $p$ parameter is higher at LDC than at NP. This could be due to the relatively thick layer of stagnant ice at LDC, which could smooth the bedrock roughness making basal sliding more likely above this layer. Around the edges of LDC, $p$ is even higher, suggesting that basal sliding is greater further from the divide. The Raymond effect (Raymond, 1983) could also be a contributing factor, making $p$ larger away from the dome. Ice may come from upstream where the glaciological conditions are different, such as different ice thickness; therefore, variations of $p$ could be an artefact of the 1D assumption. Parrenin et al. (2017) also found this trend, though their $p$ values are generally lower than those modelled in this study, especially near the LDC area. This is likely due to the addition of the variable mechanical ice thickness. It allows a layer of stagnant ice to exist which is not included in the thinning function below the mechanical ice depth. The Parrenin et al. (2017) model was artificially lowering the value of $p$ to simulate this layer. This means that the thinning function is more linear, giving a greater $p$ value for the model used in this study. The different $p$ values for the model and the ApRES at both LDC and EDC could be due to the depths of the constraints used for both fits. The deepest ApRES observations, excluding the bed reflections, are at 2145 m depth at EDC and 2017 m depth at LDC. For the 1D model, the deepest IRH constraint at EDC is 2740 m for DELORES (Table 1) and 2826 m for LDC-VHF (Table 2). Deeper constraints have a larger effect on the $p$ parameter than shallower ones due to the non-linearity of the depth–$p$ relationship increasing with depth. Therefore, the constraints which are 600–800 m deeper for the 1D numerical model strongly affect the modelled $p$ value.

Passalacqua et al. (2017) inferred basal melt rates using a model that incorporated geothermal heat flux, unlike in this study. They found similar results such as no melting across the LDC bedrock relief (Fig. 8). Around the edges of LDC and in the Concordia trench, we infer a melt rate that agrees with the trend suggested by Passalacqua et al. (2017). In Sect. 4.6, we showed the results for EDC in order to evaluate the model as we have observations to compare to. At EDC, the modelled melt rate was around 0.34 mm yr$^{-1}$, which agrees with the suggested presence of melt water found in the seismic sounding of the drill hole and Pas-

salacqua et al. (2017), who found the melt rate to be around 0.3 mm yr$^{-1}$. This is lower than the previous modelled result of $0.56 \pm 0.19$ mm yr$^{-1}$ from Parrenin et al. (2007). At the deepest dated point on the EDC ice core (3189 m), we found a modelled age (Table 5) around 200 kyr older than would be expected from the AICC2012 age–depth profile ($801 \pm 9.6$ ka; Bazin et al., 2013). The shape of the thinning function means that the age scale increases exponentially to infinity as the mechanical thickness $H_m$ is reached. Looking at the AICC2012 EDC profile determined from experimental measurements, it follows an exponential profile until the lower 200 m of dated ice, perhaps meaning that the thinning is for some reason lower than the model would expect. This effect was also observed by Obase et al. (2023, see Fig. 6 of that publication) at Dome Fuji, who used the Lliboutry vertical velocity profile (Eq. 3) in a transient 1D model. They found that the AICC2012 chronology begins to deviate from their exponential modelled age–depth profile at around 300 m above the bed at Dome Fuji, where the true age is then significantly younger than the modelled one. This could mean our age predictions for BELDC are overestimated at depths up to 300 m above the bed where the relationship becomes less exponential. Further uncertainty is added when considering the $198 \pm 44$ m modelled stagnant ice thickness at BELDC and whether it can be included in this area of reduced thinning at the base of the ice sheet.

The 1D nature of this model means that effects due to horizontal flow are not considered. Along the Dome C divide, this is a reasonable assumption. However, where ice flows along the Concordia trench as seen in the HiCARS radar dataset, the model becomes less reliable (Fig. 6a). The HiCARS radar dataset, as a result of being airborne, has a lower spatial resolution than the other datasets, and regions with faster ice flow can make IRHs difficult to trace. Over Concordia trench there appear to be some anomalies in the modelled maximum age (Fig. 11a) which are due to discontinuities in the traced IRHs. Figure 6 shows that the reliability index (Eq. 5) is less than 2 in most of the surveyed area and even less than 1 over LDC and NP; therefore, our assumptions seem to be appropriate for areas close to the dome.

In order to account for horizontal flow, we suggest that a 2.5D model would be most appropriate, as a pseudo-steady geometry could be maintained, and a Lagrangian/semi-Lagrangian tracer scheme would not be required. Sutter et al. (2021) showed that it is possible to use a non-pseudo-steady model, though it is more challenging as it requires more detailed knowledge of boundary conditions and temporal accumulation variations. Currently the spatial resolution for this type of model is also too low to take into account the scales of the processes discussed here, the small scale bedrock relief and high spatial radar resolution datasets, for example.

## 5.2 Radar dataset limitations

In Sect. 4.6, we look at the radar data sites that pass closest to EDC, which includes four DELORES transects and one LDC-VHF transect. The most accurate estimate of the ice thickness at EDC is $3273 \pm 5$ m (Parrenin et al., 2007), determined using the length of the core and the drilling cable and accounting for hole inclination. In this study, the total ice thicknesses in Table 4 come directly from the bedrock interface traced in each radar dataset and agree well with Lilien et al. (2021), who independently traced the bedrock depth in the LDC-VHF dataset as 3238 m. They suggested that the difference between the radar measured bedrock return and the drill hole bottom measured could be due to factors such as "off-nadir reflection, debris in the ice, small differences in topography over the 178 m offset, and uncertainty in firn-air content and wave speed". The DELORES dataset is also shallower than expected, probably for similar reasons.

The depth distribution of the IRHs traced in each dataset is quite different. Both the HiCARS and DELORES datasets have dated IRHs from $\sim 10$ ka ($\sim 300$ m depth). Whereas for the LDC-VHF dataset, it was only possible to trace IRHs below 1000 m because of the length of the transmission chirp ($8 \mu s$), which blanks the upper parts of the ice sheet such that the first observable IRH is much older (73.3 ka). The age of the oldest dateable IRH is limited by the radar transect linking LDC to the EDC ice core for dating. The basal melting along the divide between EDC and LDC (Fig. 8a) means that deep IRHs are either discontinuous or completely melted in some places. Therefore, IRHs over 500 ka can only be discontinuously traced from LDC to EDC, as done by Lilien et al. (2021). In this study, the same method was used to trace four further discontinuous IRHs in the DELORES radar data; however, the uncertainty was too high for them to be considered here. Cavitte et al. (2021) traced seven deep IRHs in the HiCARS dataset which could not be linked to EDC. They used the Parrenin et al. (2017) model to date the IRHs and found that, at LDC, their oldest IRH was $\sim 700$ ka. A new age–depth profile from an ice core at LDC, such as at BELDC or MYIC, would mean that deeper IRHs could be continuously traced to an ice core record for dating. Having deeper dated IRHs would help to constrain the model at lower depths, giving a more accurate thinning function, and would therefore yield more accurate model results for the whole LDC area.

The ability to observe the internal structure of the basal unit in the radar data is relatively new (Cavitte, 2017; Lilien et al., 2021), as previously radar systems did not have a sufficiently high vertical resolution, and the improvement of 2D focussing schemes with the use of multiple radar system channels has helped tremendously. As a result, no tracing convention has yet been established for the basal unit. The identification of the top of the basal unit depends strongly on the chosen convention of the human tracer, which may explain the differences between this study and that of Lilien et al. (2021). Further work should be done to establish a standard procedure for the treatment of internal layers, as for instance considered by the SCAR Action Group AntArchitecture (AntArchitecture Steering Committee, 2023) for tracing the basal unit in radargrams.

## 5.3 Nature of the stagnant unit

Tison et al. (2015) found that the palaeoclimatic signal from the ice in the bottom 60 m of the EDC ice core was not clear and the timescale distorted. They suggested that since this deepest ice was almost at the melting point, there could be a mechanism of chemical sorting acting on the impurities. Bell et al. (2011) presented radargrams which they interpreted as refrozen ice at the base of the Dome A ice sheet. The appearance of the radar data of the basal unit at Dome C is very different to Dome A, as IRH fragments are visible. However, ice with entrapment of basal debris, partly containing refrozen ice (e.g. from regelation in the past), cannot be ruled out as a source of stagnant ice, as it could be that the Dome A radar system was not able to pick up weaker, fragmented signals. The upper part of the EFZ observed by Drews et al. (2009) at EDML seems to have been a signature of the radar, related to sub-resolution changes in physical properties, as the Ruth et al. (2007) analysis of the ice core showed that a palaeoclimatic signal could be found in the top half of the EFZ. Cavitte (2017) trace what they call "deep scattering zones" in the HiCARS dataset, as well as a single profile flying across Dome C (see their Fig. 5.17), showing furthermore that their modelled IRH geometries do not match those of the deepest traced IRHs, suggesting a stagnant basal layer. The basal unit observed in the LDC-VHF data of this study and published in Lilien et al. (2021) is different because the radar system is sufficiently sensitive at this depth, so the effect is likely to come from physical changes in the ice. The ApRES measurements at LDC show that the ice at the depth of the basal unit has different properties to that above. Moreover, the magnitude of the ApRES-LDC vertical velocity (Fig. 3a) decreases to zero more quickly than at EDC (Fig. 3b) at similar normalised depths. It follows from the ApRES-LDC measurements that there is almost no deformation in the basal ice, so it must be nearly or totally stagnant. Our model effectively shows that in order to better fit the IRHs the vertical strain rates near the surface must be higher over LDC than would be produced by a profile which follows the Lliboutry (1979) variation over the full depth.

In Sect. 4.3 we showed that the thicknesses of the modelled stagnant ice and the observed basal unit are comparable and have similar shapes. There was a systematic offset where the average difference between the modelled and observed basal layers was 40 m. This could be a product of some of the issues we have already discussed, such as the difficulty in tracing the radar basal unit or the suitability of the Lliboutry thinning profile in the deepest part of the ice sheet. However, the ApRES results also showed that the best fit for the vertical

velocity profile occurs when we include a layer of stagnant basal ice. Therefore our results are compatible with the hypothesis that this basal unit observed by the LDC-VHF radar system is indeed stagnant. It appears that irregularities, such as local dips, in the bedrock are filled in by the basal layer. The IRHs therefore follow the smooth shape of the top of the basal layer, which could have implications for the overall ice sheet dynamics. For example, if the ice sheet is sliding above the basal unit, it could smooth the roughness and decrease basal drag. This would in turn increase the horizontal flux. However, the thickness of the flowing ice decreases, which would reduce the total horizontal flux. In order to quantify which of these mechanisms has a larger effect, flow modelling in different areas of Antarctica would be required. We note here that although the ApRES measurements indicate that ice is vertically stagnant, that does not rule out the possibility of horizontal advection. Our age model, with the reliability index, and the characteristics of the area – slow flow, relatively shallow thickness with no sign of subglacial melting – suggest that horizontal advection has a small role in this particular location. There are many unanswered questions regarding the basal layer. It is unknown if the basal layer exists elsewhere in Antarctica and, if so, how much of the ice sheet it covers. If it is widespread, perhaps the resultant decrease in basal drag could be incorporated into future sea level estimates.

The ice core to be extracted at BELDC could yield a basal layer almost 200 m thick. Analysing its fabric structure, age, air composition, isotopic composition and impurities will substantially advance our understanding of the basal layer observed around Dome C.

## 5.4 Oldest Ice prospects

We confirm that the BELDC site is promising for retrieving ice old enough to study the MPT. Our results for BELDC are compatible with those of Lilien et al. (2021), who found that the 1.5 Ma isochrone is at $2498 \pm 14$ m depth at BELDC with an age density of $19 \pm 2$ kyr m$^{-1}$. Our results therefore confirm that, despite the stagnant ice present in the bottom of the ice sheet, the BELDC drill site should reach its target of 1.5 Ma, although it might be at the limit in terms of acceptable age density. It is also unclear to what depth we can trust the exponential age–depth profile predicted by the Lliboutry assumption (Eq. 3). A safer expectation for Oldest Ice is 1.2 Ma, as the modelled age density is 12.7 kyr m$^{-1}$, which is well within the 20 kyr m$^{-1}$ target (Fischer et al., 2013). Also, it should still cover the MPT and the last part of the 40 kyr world. The MYIC site has very similar characteristics to BELDC. The differences between the two ice cores could help inform future site prospecting, as they may show evidence of processes that are not detectable from the radar data or have not been considered in the model.

We also investigated NP as a potential area for finding old ice. According to the model, age density at 1.5 Ma is twice as good as that of LDC. This is due to the relative lack of stagnant ice predicted by the model at NP, which at LDC causes more thinning in the deepest ice. The reason that NP was not considered further for the Beyond EPICA project was that basal melting at LDC was much less likely, making it the safer option. While the NP was initially flagged by a HiCARS transect passing nearby, the only detailed radar data available at NP are the DELORES transects. In order to determine the suitability of NP as an Oldest Ice site, it would be useful to conduct a radar survey using the multichannel coherent radar depth sounders such as the LDC-VHF. The methodology implemented in this study can be applied to other areas of Antarctica in the search for Oldest Ice such as Dome A, Dome B and Dome Fuji (Wang et al., 2023).

## 6 Conclusions

In this paper, we presented a 1D numerical model which interpolates and extrapolates the age–depth profile obtained from radar IRHs. The model was applied to three radar datasets collected over the LDC area and was shown to be more relevant when a layer of modelled stagnant ice could be included. Model outputs show similar results for the three radar datasets regarding the stagnant ice thickness and maximum age in the lowest part of the ice. It was shown that the thickness of the modelled stagnant ice is comparable to that of the basal unit observed in the radar, supporting the hypothesis that this basal unit is stagnant. ApRES measurements at LDC also showed that the vertical velocity profile can be best explained when a basal layer of stagnant ice is present. The maximum modelled age at an age density of 20 kyr m$^{-1}$ for the transect location closest to the BELDC drill site is $1.45 \pm 0.16$ Ma at $2494 \pm 30$ m depth. The resolution of 1.2 Ma old ice is 12.7 kyr m$^{-1}$ at a depth of $2478 \pm 30$ m. The predicted thickness of stagnant ice at the base of the ice sheet is $198 \pm 44$ m. The MYIC site on LDC also yields similar modelled results. The model was also applied to NP, 10–15 km north of EDC, where the maximum age was found to be around 2 Ma, and the age density of 1.5 Ma ice was around twice as good as that found at the LDC drill sites, making it a potentially promising site for future Oldest Ice projects. The 1D model can be applied to any other area that is close to a divide so that horizontal flow is negligible. The development of a more complex 2.5D model which takes into account the horizontal flow of an ice sheet will allow us to look for old ice in other areas of Antarctica, especially with the collection of new, more spatially extensive and denser radar campaigns.

*Code availability.* The code for the 1D numerical model is available publicly from https://github.com/ailsachung/IsoInv1D (last access: 7 August 2023; DOI:

https://doi.org/10.5281/zenodo.8189792, Chung and Parrenin, 2023).

*Data availability.* The HiCARS IRHs used in this study can be found publicly on the US Antarctic Program Data Center (USAP-DC): https://doi.org/10.15784/601411 (Cavitte et al., 2020). The DELORES profiles discussed in this paper will be available from https://data.bas.ac.uk (last access: 7 August 2023). The LDC-VHF radar will be made publicly available in a PANGAEA repository. The LDC-VHF IRHs used in this study are available here, https://doi.pangaea.de/10.1594/PANGAEA.957176 (Chung et al., 2023), and the DELORES IRHs will be made available in a public PANGAEA repository. The BedMachine version 3 bed elevation data come from the work of Morlighem et al. (2020) and are available on https://nsidc.org/data/NSIDC-0756/versions/3. The REMA surface elevation data are from the work of Howat et al. (2019), available on https://www.pgc.umn.edu/data/rema/.

*Supplement.* The supplement related to this article is available online at: https://doi.org/10.5194/tc-17-1-2023-supplement.

*Author contributions.* AC traced IRHs in the DELORES and LDC-VHF datasets with help from RM, DS and OE. MGPC provided the HiCARS IRH inputs. AC also improved on the model, developed by FP with input from MGPC, and ran experiments with supervision from FP and OE. DS, RM, DT, DAL, VH, CR, CM, CO, MF, PG, HM, DDJ and OE were involved in the DELORES and LDC-VHF radar survey design, data acquisition and processing. AC prepared the paper with relevant input from all co-authors.

*Competing interests.* At least one of the (co-)authors is a member of the editorial board of *The Cryosphere*. The peer-review process was guided by an independent editor, and the authors also have no other competing interests to declare.

ons expressed and arguments employed herein do not necessarily reflect the official views of the European Union funding agency or other national funding bodies.

*Special issue statement.* This article is part of the special issue "Oldest Ice: finding and interpreting climate proxies in ice older than 700 000 years (TC/CP/ESSD inter-journal SI)". It is not associated with a conference.

*Acknowledgements.* This publication was generated in the frame of Beyond EPICA. The project has received funding from the European Union's Horizon 2020 research and innovation programme under grant agreement no. 815384 (Oldest Ice Core). It is supported by national partners and funding agencies in Belgium, Denmark, France, Germany, Italy, Norway, Sweden, Switzerland, The Netherlands and the United Kingdom. Logistic support is mainly provided by ENEA and IPEV through the Concordia Station system. This publication was generated in the frame of the DEEP-ICE project. The project has received funding from the European Union's Horizon 2020 research and innovation programme under the Marie Sklodowska-Curie grant agreement no. 955750. AWI acknowledges Pascal Andreas and Sven Lenius for their contribution to the preprocessing of the data. This is Beyond EPICA publication number 32. Marie Cavitte is a postdoctoral researcher at the F.R.S-FNRS.

*Financial support.* This research has been supported by the Horizon 2020 research and innovation programme (grant nos. 955750 and 815384).

*Review statement.* This paper was edited by Joseph MacGregor and reviewed by Johannes Sutter and one anonymous referee.

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

**Remarks from the typesetter**

TS1    Please give an explanation of why this needs to be changed. We have to ask the handling editor for approval. Thanks.