# Peer review of "Figure S1. Modelled age depth profile at the BELDC drill site along with age uncertainty. Results for the DELORES dataset are in orange, results for the LDC-VHF dataset are in blue with the shaded areas showing $1\sigma$ uncertainty. Black circles are the isochrone ages with their age uncertai"

_EGUsphere, 2023_

## Author Comment (AC1)

**Stagnant ice and age modelling in the Dome C region, Antarctica**
**Review 1 response**

**We thank the reviewer for the time and effort for evaluating our manuscript. Below, our response is in blue and changed text in the manuscript in blue italic:**

Reviewer comments
**Author comments**
*Additional/modified text passages from manuscript*

Review of "Stagnant ice and age modelling in the Dome C region, Antarctica" by Aildasa Chung et al. This paper examines the age of the ice interior of Dome C using a 1D ice flow model combined with radar imagery. Chapter 2 describes the 1D ice flow model based on Parrenin et al. (2017) with the incorporation of mechanical ice thickness and stagnant ice. A method for optimizing unknown parameters (precipitation, flow parameters, and mechanical ice thickness) using the ages from radar imagery are described. Chapter 3 describes a method for detecting basal units and age layers from radar images, and their correspondence to the age profile from the EDC ice core. In Chapter 4, the 1D model results are validated against the age and vertical velocity profiles of the LDC or EDC, and the correspondence between the stagnant ice distribution. The results of the 1D model regarding spatial distribution of stagnant ice are compared with radar images. Chapter 5 examines uncertainties from the ice flow model and radar datasets.

Overall, I think the paper is of sufficient quality to be accepted. Below are some questions and suggestions for minor revisions.

**We have rearranged the results section following suggestions from another reviewer. This rearrangement has changed figure numbers, sections number and line numbers. We have tried include new numbers in our response so it is easier to find the revisions.**

**We first present evidence from observations of the stagnant basal layer – the ApRES vertical velocity profiles at EDC and LDC, then the basal unit seen in the LDC-VHF radar survey. We then compare our numerical model with and without inverted bedrock depth, in order to show that a model which allows for a stagnant ice layer is more appropriate, at least in the Little Dome C region. Having shown that the stagnant ice model is more appropriate, we then present the melting/stagnant ice, p parameter, accumulation rate, age and age density results for all four areas of interest around Dome C. Finally, we present the results at single locations- EDC for comparison to experimental ice core data, and LDC predictions for the current Beyond EPICA and MYIC drill sites.**

L43: I understand that "stagnant ice" refers to ice masses with a minimal flow. Meanwhile, I think it would be meaningful to describe a definition of "stagnant ice" in this study.
**Definition added new line 79:**

*We label this ice "stagnant" as the best fit thinning profile of the 1D model does not pass below $H_m$, though from observations we can see that ice continues to depth Hobs, so we infer a vertical ice flow velocity of 0 for this layer.*

L80: Is r(t) exactly the same as in Figure 2 of Parrenin et al. (2017)? If so, I recommend citing the figure.
**Suggestion accepted**

L83: "temporally-averaged" accumulation? And, is it averaged over the last 800,000 years?
**New line 86 changed to:**
*temporally averaged value over the last 800 ka*

L87; Actual basal melting should be determined thermodynamic, so I think this formulation is one assumption. Does this formulation come from a condition of no discontinuity in the vertical velocity at the observed bedrock?
**Basal melting can be determined thermodynamically and kinematically. We opt for the kinematic approach, truncating the vertical velocity at the observed bedrock.**

L90: Name of the software?
**New line 98:**
*For this model, we use the Python module SciPy's least-squares optimisation with the Trust Region Reflective algorithm.*

Equation 5: What is the definition of σiso? And also, write out the term "reliability index" in the description of equation 5 as the term is used later (Figure 12 and Section 5)
**New line 104:**
*The depths and ages of isochrones are diso respctively and χiso, σiso is the age uncertainty and χmod is the modelled age.*
**New line 110:**
*If the model is a good fit, then the "reliability index" σR is close to 0.*

L110: Any introduction for MYIC?
**MYIC now mentioned in the introduction new line 28:**
*The Australian Antarctic Division (AAD) have also selected a drill site at LDC for their Million Year Ice Core (MYIC) project.*

Table 3: "DC-LDCRAID2", "DC_LDCRAID", "DC_LDC_DIVIDE", and "DC_PNV09B" are not mentioned in the text. Which panel in Figure 2 does these names correspond to?
**Radar line names have now been removed from Table III as they did not add any useful information for the reader. The radar lines are not fully visible in Fig. 2 as we focus on LDC where the majority of the radar survey took place for panels (b) and (c).**

L203: Total ice thickness at EDC?
**New line 315: We are referring to the total ice thickness at the point in the radar closest to EDC.**
*The closest point to EDC in the LDC-VHF dataset is 178 m away with a total ice thickness of 3239 m.*

L208: High melting area in the lower left of the figure may not be reliable, according to Figure 12. It may be hard to explain why there's considerable basal melting where the bedrock elevation is relatively high.
**New line 286, line added to reference former fig 12 (now Fig 6)**:
*There is significant melting predicted around the edges of the LDC relief, especially on the western side of LDC and across the Concordia trench (Fig. 2a), where Fig. 6a shows that the model is less reliable*

Figure 5: Where does this transect correspond (on the map)?
**Now Fig. 7: Labels A and B now show transect on the map and the caption has been extended to describe this.**
*A radar transect in the LDC-VHF dataset which passes diagonally across Patch B from A to B in panel (b).*

Figure 6: The caption in Figure 6b would be "p=3.6, and stagnant ice=0" based on sentences.
**Suggestion accepted**

L272: Confused, because according to figure 6, the p=3 for LDC. Why does figure 8 have a more significant value of p? This may come from different radar/ApRES velocity measurement datasets. Please discuss this.
**Former Fig. 8 (now Fig. 10) shows the modelled p parameter which for LDC is higher than the ApRES fit at LDC - former Fig 6a (now Fig. 3a).**
**Now added to the discussion:**
*The different p values for the model and the ApRES at both LDC and EDC could be due to the depths of the constraints used for both fits. The maximum ApRES measurements, excluding the basal reflections, are 2145 m at EDC and 2017 m at LDC. For the 1D model, the deepest isochrone constraint at EDC is 2740 m for DELORES (now Table II) and 2826 m for LDC-VHF (now Table III). Deeper constraints have a larger effect on the p parameter than shallower ones. Therefore, the constraints which are 600-800 m deeper for the 1D numerical model, strongly affect the modelled p value.*

Figure 7: High precipitation areas in the upper left corner might be less reliable, according to Figure 12.
**New Fig. 9. Surface accumulation is mainly constrained by the sallowest isochrones, so it should be a robust feature even if the reliability index is not so good.**

Table 4: What are the values of p and a in these modeling results?
**p and a have now been added (now Table V)**

L325 For this discussion, I think it's necessary to refer to Parrenin et al. (207) (Equations 4-5), which discusses the relationship between basal deformation and the value of p

**When Lliboutry (1979) developed their numerical scheme, p was initially calculated depending on the vertical temperature gradient and ice sheet thickness (Eqs. 4 and 5 of Parrenin et al. 2007). However, this theoretical value of p is not compatible with the mechanics of a divide and does not take into account basal sliding. Therefore, for our case at LDC, the best course of action for us is to invert p as there are too many unknowns to calculate it.**

---

## Author Comment (AC2)

**Stagnant ice and age modelling in the Dome C region, Antarctica**
**Review 2 response**

**We thank the reviewer for the time and effort for evaluating our manuscript. Below, our response is in blue and changed text in the manuscript in blue italic:**

Reviewer comments
**Author comments**
*Additional/modified text passages from manuscript*

General Comments

Chung et al. present an interesting study investigating the age-depth profile around the oldest ice site Little Dome C and the potential future site North Patch using a 1-D model. They include the possibility of stagnant ice in the model as a free parameter to better match the observed isochronal scaffold and predict the oldest ice in the area. The presence of such a layer is suggested by radar data, especially by vertical velocity observations inferred from ApRES data. The study is well written and will make a valuable contribution to the oldest ice canon. What makes it especially appealing and exiting, is the fact that hopefully in a couple of drilling seasons the modelling assumptions and findings can be confirmed or rejected.

I do have some concerns which should be addressed before publication. None of these comments are major issues, but rather address the way in which the methods, results and uncertainties are presented.

1: A more detailed discussion of the uncertainties is in order. While the authors provide a section on modelling limitations it is rather general and does not really include quantitative statements. This especially pertains to the issue of the exponential age-depth profile close to the bedrock (or alternatively close to the stagnant ice boundary). The authors do provide a wide list of factors contributing to uncertainty but I suggest including these (and quantify where possible) more prominent in the results sections already.

**1: Quantifying model uncertainties**
**We are not able to quantify contributions to uncertainty without comparison to a more complex model which incorporates a layer of stagnant ice. To our knowledge, such a model does not yet exist therefore we can only qualitatively discuss the factors which contribute to uncertainty. We do compare our results to both ApRES measurements and the basal unit observed in the radargrams and we quantify the difference to give an idea of the accuracy of the model. (See below for comment on the results/discussion structure). In addition, the fact that the basal unit observed from the profiling radar data also coincides with the modelled layer of stagnant ice at least provides confidence in the results, although we cannot provide a full quantification unless we consider the difference in observed and modelled depth of the basal unit top as a metric.**

2: A more detailed description of the 1d model and assumption flowing into it would be in order. I know that it has been described extensively in previous publications but a little more detail would be nice. Especially given the fact, that the 1d modelling including a stagnant ice layer is the major focus of this study.

**2: more model details**
**We appreciate the interest of the reviewer in the model and are happy to add more detail on the 1D model as it is the major method in our study but we consider putting it in the supplement so we don't lose readers who are less interested in the model details.**
**Additional text added to supplement:**
*1D numerical model details*
*We use a modified version of the 1D numerical model in Lilien et al. (2021). We integrate over 1/τ (Eq. 3) with linear interpolation between vertical nodes. The thinning function is fairly linear near the surface and changes more quickly with depth towards the bedrock, therefore we use a quadratic vertical grid. This means that the nodes are spaced further apart near the surface and are closer together near the bedrock. We use an average step size dζ = 0.001 and a ratio of 0.1 meaning that the distance between the first node at the surface and the second below is 1.9dζ. The distance between the penultimate node and the bedrock node is 0.1dζ.*

Currently, there is much more emphasis on the description of the radar systems and field seasons, which is nice but they have been discussed in the original publications. You could save some space there and in turn expand the model description/physical reasoning behind a stagnant ice layer.

**2: detail on radar systems**
**Regarding the LDC-VHF survey, only a single line was discussed by Lilien et al, not the complete survey. This is the first publication of these data, so the description has to be more comprehensive. We do also have to mention that the conversion to depth from twt for HiCARS and DELORES is different to Cavitte et al. 2021. We also included some details of the radar systems used by Cavitte et al. 2021 so that all three radar survey datasets in section 3 are described with a similar level of detail.**

3:For the reader to appreciate the advance represented in introducing a stagnant ice column Hm it would be nice to have a comparison to 1d modelling where this column is not assumed. I can see that adding a free parameter which you can optimize leads to a better fit with the traced and dated internals. However, I am missing

- a physical explanation of why there should be a stagnant ice column. I see that in section 5.3 you discuss the nature of the stagnant ice, but this is almost at the end of the paper. I suggest to introduce this in a concise manner earlier
- a quantification how this assumption improves the fit over the previous version of the 1d model.

I don't know how expensive it is to run the previous version over the transects. You could also select a few points for a comparison. How much do we gain by assuming a stagnant ice column, how does the age profile look like, if you don't assume it. Your current Figure 11 could be also done with no inclusion of a basal unit (either in an additional figure or included in the same figure).

**3: model "no stagnant ice assumption"**

**We thank the reviewer for this suggestion and we agree that having this comparison improves our study. We have now run the model with the no stagnant ice assumption. In fact, this has been done extensively during the pre-site survey phase of Beyond EPICA, until we found the need to include a stagnant ice layer. One advantage of this model is that the computation time is relatively low - transects from all radar datasets took around 2 hrs total to run. We now show a comparison of the two models using the Basyian Information Criterion (BIC) which is a measure of the relevance of one model over another. We now show the age profile assuming no stagnant ice at BELDC in Fig. 13 (formerly Fig. 11) and at MYIC (Fig. S1), where it is clear that the stagnant ice model provides a better fit to the isochrone constraints.**

Structure: there is a lot of information especially in figures with maximum age/age density etc along the radar transects. As a reader I think the central figures are figure 6 (ApRES derived velocity which motivates the assumption of stagnant ice) and figure 11 (the actual age depth profile at BELDC). Figure 6 could follow right after Figure 1 actually as it provides the physical data supporting the assumption of stagnant ice. Some streamlining of the sections would further improve accessibility.

**Structure**
**We have taken the reviewer's comment into consideration and have now rearranged the results as follows.**

**We first present evidence from observations of the stagnant basal layer – the ApRES vertical velocity profiles at EDC and LDC, then the basal unit seen in the LDC-VHF radar survey. We then compare our numerical model with and without inverted bedrock depth, in order to show that a model which allows for a stagnant ice layer is more appropriate, at least in the Dome C region. Having shown that the stagnant ice model is more appropriate, we compare results directly with the observations - ApRES and basal unit. We then present the melting/stagnant ice, p parameter, accumulation rate, age and age density results for all four areas of interest around Dome C. Finally, we focus on the results at single locations of interest- EDC for comparison to experimental ice core data, and LDC predictions for the current Beyond EPICA and MYIC drill sites.**

**A note: we structured the paper so that the results section contains purely the numerical results and maps. Then in the discussion, we detail our interpretation of the results and sources of uncertainty. This structure disagrees with a few of reviewer 2's comments as they would like more interpretation in the results sections as they find that the stagnant ice aspect gets lost a bit in-between the model results. However, by presenting the observations first as motivation, then adding the comparison of models with and without the stagnant ice assumption, we draw the focus again to the fact that including modelled stagnant ice is more consistent with observations which is one of the main points of the paper.**

Minor comments:

Very verbose abstract with technical details
e.g. :… here defined as the age at a maximum age density of 20 kyr m-1
**Removed and details included with first sentence suggested below**

What do you mean by 'seem'? Is this uncertain due to measurement uncertainties? If not I'd suggest to drop 'seem'
**Suggestion accepted**

Suggest to make this the first sentence and then motivate why:
The European Beyond EPICA project aims to extract a continuous ice core of up to 1.5 Ma, with a maximum age density of 20 kyr m−1 at this site called Beyond EPICA Little Dome C (BELDC).
**Suggestion accepted**

l27 ... is whether the deepest ice lying just above the bedrock proves useful for paleoclimate reconstructions?
**Suggestion accepted**

L30 submeter?
**Suggestion accepted**

L31 is this published? If there is a theory behind why I suggest to quickly mention it here.
**This refers to Tison et al. 2015. This part has been reworded as below, to make the reference clear.**
**New lines 34-38:**
*Generally known as the basal layer, the mechanics of this deformed ice are not well understood. At the bottom of the EDC ice core, there is a section of around 60-70 m where Tison et al. (2015) found that the paleoclimatic signal had been disturbed, perhaps due to a chemical sorting mechanism cause by the ice being close to melting point. While the isotopic composition of this ice was studied by Tison et al. (2015), the interpretation of these results remains challenging. The mechanical stress on the deepest ice has distorted the timescale and left no continuous record.*

L33 what do you mean by 'uncertain'? suggestion : interpreting these results remains challenging
**Suggestion accepted**

L34 no continuous record?
**Suggestion accpeted**

L35 is it not called echo-free zone anymore, should it be called differently? Sorry for my ignorance, I am no radar expert.
**As it turned out not to be a physical property of the ice, but an artifact of the radar, the EFZ cannot really be thought of as a "basal layer" of ice anymore. (Technically, the zone presented in the Drews et al (2009) radar could still be called the EFZ because it is "echo-free" but covers an undisturbed section of ice (with an intact paleoclimatic signal) as well as a disturbed section, where paleoclimate information cannot be retrieved in an ice core.)**
**New line 38:**
*Basal ice was difficult to observe using previous radar systems due to the presence of an echo-free zone (EFZ).*

I have the feeling the introduction could be rearranged somewhat. As of now, it jumps from topic to topic making it a bit strenuous to follow.
**We think that the current order of the introduction makes most sense as we start with motivation and move to past research on basal layers generally, then finish with the most recent modelling and observations in the Dome C region – our area of interest.**

L47: be caused by a backscatter power that is sufficiently far below the noise level and therefore …
**Suggestion accepted**

L50 :

So Lilien et al. used a 1d model (different approach?) already. I would thus rephrase the subsequent sentence, simply stating that your objective is now to expand this analysis to the whole Dome C region (with a different, more robust?, method)  and not stating that this would give a better idea (this sort of diminishes Lilien et al.'s work).

Suggestion: Lilien et al. (2021) accessed the age-depth profile at the BELDIC site, inverting the optimal value of the thickness of a layer of stagnant ice, which was found to be close to the observed thickness of the basal unit. Here, we expand on their work investigating the whole Dome C region presenting (is this the first time this approach is presented? If not, use e.g. 'employing' and cite) a 1D numerical model which uses inverse methods to infer a layer of stagnant ice from the isochronal information. This approach will elucidate the spatial extent of this inferred stagnant ice layer and its impact on the age profile in the region.
**The model is slightly modified from Lilien et al. 2019, details to be put as mentioned above. But the base idea is the same.**
**Changed to:**
 *Lilien et al. (2021) found that at the Beyond EPICA drill site onoptimal value of the thickness of a layer of stagnant ice, which was found to be close to the observed thickness of the basal unit. Here, we develop the idea further by investigating the whole Dome C region using a similar model to Lilien et al. (2021) but with a different numerical scheme and optimization method (see supplementary material). This approach will elucidate the spatial extent of this inferred stagnant ice layer and its impact on the age profile in the region*

I assume that you don't capture ice-dynamic behaviour in a 1d model?
**We are unsure which ice-dynamic behaviour you are referring to.**
**The p parameter is inverted in our model and from it is possible to infer basal sliding for example, it is this ice-dynamic behaviour that we refer to, see discussion on modelling limitations (Sec. 5.1). We also suggest a 2.5D model could be used to study some aspects of ice dynamic behaviour.  In terms of temporal variations, ice thickness or dome position for example, these quantities cannot be inferred from a pseudo-steady model.**

Parrenin et al use the assumption of covariance between melting and surface accumulation to use a analytical expression of the thinning function and state that this only leads to an error of <6% in the thinning function. How would this error propagate in the method applied here, what does it mean for the age uncertainty? Generally, I would suggest to expand the discussion of uncertainties due to the assumptions made in this study.
**We refer to our general comment above, that it is difficult to quantify our uncertainties as there is currently no more complex model which we can use for comparison.**

L105 which were taken during the period … and informed the selection …
**Changed to *"…which were taken during the period of the 2016-2020 Antarctic field seasons and informed the selection…"***

L113 …assuming an electromagnetic wave velocity in ice of … as in Winter et al. … [I assume this is not a universal number, but a deliberate choice]
**The value presented in Winter et al. is the most recent value, determined from combining various radar systems with ice-core dielectric properties, which has been used in previous studies with the same radar systems (Cavitte et al. 2021 and Lilien et al. 2021).**

L114-116 maybe mention here shortly why you briefly describe this.
**New line 136, sentence added.**
*These measurements give us an insight into the internal ice deformation, offering further evidence of a potential stagnant ice layer.*

While I appreciate the description of the different radar setups and campaigns including uncertainty assessments, I have the feeling that they are quiet extensive compared to the description of the 1d model, the assumptions flowing into it and the corresponding uncertainties. I would therefore extend the discussion of the 1d model a little (which barely covers a single page) and trim down the radar setup description (right now 4 pages!).
**Please see our response to the general comments above. We moreover want to clarify that the previous publication of Cavitte et al (2021) only used part of the DELORES data set (<20 lines) while we now reprocessed and analysed the full set of 120 lines. We consider that this warrants to describe the data in the current way, especially also as they will be made available as open data to the public. The current description will make it easier for later users to adequately use them.**

I recommend this especially considering that in the introduction you note that you present a 1d model. So the reader would assume the modelling is the focus here and not the field work/equipment/technical aspects which have been described in detail elsewhere.
**This is a typical manuscript where the results are produced by the application of a model, which is driven by observational data. We agree that it should be clarified that this sort of "assimilation" type approach should give more weight to the model, but as explained above we do not think that the radar observations are minor – especially as their area-wide presentation is crucial for understanding the age-depth distribution at both drill sites for later interpretation. We rather consider that in the past too much focus has been put on pure model results and observations were given too little weight. Moreover, as laid out in the overall Beyond EPICA objectives, the observations and analyses also serve as a template for other projects targeting very deep and/or old ice, making a more extensive description mandatory. Often we are asked in reviews to explain the data recording and processing in more detail so that observational scientists can copy the process, so it seems to depend on the reviewer's own background how people consider the balance. We tried to shift the manuscript to a new balance in the revision.**

Many of the things you list in your radar/fieldwork description could be neatly summarized in a table (number of IRHs, depths, ages, coverage, no of transects etc.). This would make it much more accessible.
**Thank you for your suggestion. We added a table to make this information more clearly visible.**

Confused by the section header: Inferred ages for EDC
**Changed to *"Model results at EDC"***

What you discuss here are modelled ages in closest vicinity to the EDC-site (closest point on transect)? Maybe the header should reflect this (likewise for tables 1 and 2 which say age at EDC, should prob read closest point to EDC as in Table 3).

**Tables I and II (now II and III) now read *"Depth nearest EDC"* with the caption explaining that this is the closest point on the radar transect.**

**Table III (now Table IV) reads *"distance of closest point to EDC"***

I assume this is to give an idea for the estimated age variation around EDC I was not sure what to take away from this rather compact subsection. You propose to evaluate the accuracy of the model. To me this section suggests that there are high age variations (~200 ka) around the EDC site. Or are they assumed to be uniform and thus the 200 ka variations are a measure of uncertainty of the model?

**We refer to our general comment on structure. In the results section we detail numerical results, then in the discussion (sec 5.1 model limitations) we compare to literature expectations for the EDC, discuss sources of uncertainty and possible reasons for disagreement.**

Also, how do I interpret ages at 3189 m depth if the the total ice thickness at the respective locations is <3189 m (as is the case for DC_PNV09B)? Maybe I misunderstand?

**This age has now been removed from the table as it can be calculated by the model but you are correct- there is no physical meaning in this case.**

Section 4.2

**New sec 4.4**

Here the model results for the stagnant ice column are discussed. As the assumption of a stagnant ice column is the main focus of this paper I would suggest you expand a little on that and remind the reader of the implications of the modelling/inversion/optimisation exercise. It is otherwise really easy to miss the main focus of the paper.

**As we have now added a section on model comparison with and without the stagnant ice assumption (Sec. 4.3), we bring the focus back to the modelling after discussing observations of ApRES (4.1) and basal unit (4.2).**

**New line 283:**

**We have also added an introductory sentence to New Sec. 4.4 (former sec. 4.2) to remind the reader.**

*Our model uses an inverted mechanical ice thickness Hm (Fig 2) to infer either a basal melt rate or a layer of stagnant ice.*

L204 age uncertainty for the oldest ice at dome c of pm 96 ka? Typo I assume. 9.6ka (see figure 6 Bazin et al. 2013 and supplements)?

**Yes, corrected**

L219 colormaps are inconsistent with the colorbars (see comments on Figure 3 below). I assume the red and blue colormaps are combined in the figure.

**Please see response to Fig. 3 comment below.**

In 4.2 and figure 3 you discuss/show the modelled stagnant ice/melting but then you discuss the radar derived stagnant ice column. Easy for the reader to confuse modelled and observed numbers here, as figure 3 only shows modelled quantities.

**All radar observations are now discussed before this section (now in sections 4.1 and 4.2) so it is clear that we are now presenting modelled quantities from section 4.3 onwards in the results. Section 4.4 (formerly 4.2) has now been renamed "Modelled stagnant ice and melting" to make the distinction clear.**

L221 I suggest you mention the melt rate uncertainty here.
**New line 298:**
*"This low rate is not significant relative to its uncertainty which is between 0.05-0.1 mm yr−1"*

L234: To the naked eye panel a and b are quasi identical, maybe consider plotting the mismatch between the modelled and radar inferred $H_m$ (observations in panel a, delta in panel b).
**Figure 7b (former Fig. 5b) shows the mismatch.**

L235 Figure 5a shows the modelled age of a single …
**Suggestion accepted**

L237 This is probably the strongest modelling case for a stagnant ice unit shown in the paper. It would be very nice to have a comparison for Dome C, where we have a very good age-model. What happens if you apply your 1d model including the mechanical ice thickness as an optimization parameter. Does your model suggest a stagnant layer for EDC, EDML, Dome Fuji etc.? This goes back to my general point, that a comparison to model output without the inclusion of mechanical ice thickness would very much strengthen the message of this study.
**We cannot compare stagnant ice thickness at EDC because there is none predicted by the 1D model or observed in the radar surveys or ApRES. There was melting found at the bottom when the EDC ice core was drilled and our model predicts melting there. This is mentioned in new sec 4.4 Modelled stagnant ice and melting and discussed in sec 5.1 Modelling limitations where we compare to literature values. Dome Fuji was modelled using a separate radar dataset which covered a much larger area but at lower resolution (see Wang et al, preprint,** https://doi.org/10.5194/tc-2023-35**). The 1D model is not appropriate for EDML due to the horizontal ice flow. Although in future work we are developing a 2.5D model which could be applied to areas such as EDML as we very briefly mention in sec 5.1 modelling limitations.**

L243 maybe replace 'described' by 'constrained' but maybe I am not completely clear what you mean here.
**Sentence now removed due to rearranging of results sections**

L244 this is a key sentence, but I am missing the underlying data. Where is the comparison between the fit with and without the option of a stagnant base layer?
**The comparison between the model fits using the delta BIC value is now in Sec 4.3.**

I am writing this as I am reading, so maybe this will pop up further below. Looking at Fig. 11 it seems to me that the fit in Lilien et al. 2021 which does not include $H_m$ is very good already. Surely the standard

deviation becomes smaller with an additional tuning parameter, but I would argue that this alone is not yet a convincing statement without a physical explanation as to why such a layer would be present.

**There is a stagnant ice layer present in Lilien et al.'s modeling work which is now present in Fig 13 (former fig 11) which is close to the LDC-VHF line in our work. We have now added the modelled result assuming no stagnant ice layer- note that we don't include uncertainties as we believe the deviation from the isochrone constraints alone is a clear argument against this model.**

**Sentence added new line 351:**

*For comparison, we show the age-depth profile determined using the model which does not allow for a stagnant ice layer (black line). The profile clearly deviates from the isochrone constraints (blue circles) at depths >2200 m, supporting the conclusion that the inclusion of a stagnant ice layer in the model is the more appropriate at LDC (Sec. 4.3).*

[...]

Having looked at Figure 6 now, I guess the comparison of the ApRES and modelled velocities is the main argument for the presence of a stagnant ice layer. Maybe it makes sense to show this right at the beginning? For me it is difficult how big the difference in vertical deformation is between p=3.04 and p=3.6 is. How big is p for EDC and LDC if you use the old version of the 1d model (without optimising for $H_m$)? For the readers not familiar with the peculiarities of the model it is difficult to assess the significance here.

**Since we have now introduced the BIC value, we have assessed the suitability of the stagnant ice model and shown that it is more appropriate for the dome C areas at least. We considered adding the non-stagnant ice model results to Fig. 3 (ApRES former Fig. 6). However, since we have shown that the non-stagnant ice model is sub-optimal, we think this may confuse the reader and make the figure too busy. We show the reviwer the figure below which is the ApRES measuremnts at LDC (Fig 3a) including the non-stagnant ice assumption (in yellow).**

[Figure]

**At first glance it may appear that the model fit is better with no stagnant ice, however we should consider our other analyses such as the BIC value for model relevance (Fig. 5) which show that the stagnant ice model is more appropriate for fitting to the radar isochrones. Fig 13 shows the importance of the stagnant ice model for fitting isochrones deeper than 2200 m. As there are no ApRES measurements at this depth this offers a possible reason for the different fit provided by the 1D model and ApRES measurements.**

L260 in Figure 6b it looks like the vertical velocity uncertainty (top 2000m) is smaller at the EDC site and not at LDC??
**This is normalized vertical velocity as we are mainly interested in the shape. The absolute values are larger at EDC as you would expect.**

Figure 6b I assume there is a mistake in the numbers for p as in the caption you mention that p=3.6 and no stagnant ice for EDC.
**Yes there was a labelling mistake in the figure legend, this has now been fixed.**

L262 comparing the vertical velocities at LDC and EDC at around 2000m the difference seems to be very small (maybe 0.1-0.2 m/a?). I don't know much about ApRES derived velocity uncertainties, but this seems to be somewhat narrow margins? Again, a more expansive discussion of uncertainties and potential alternative explanations would help a lot here.
**We follow the standard ApRES uncertainty estimation technique described in Brennan et al. (2014) and Nicholls et al. (2015), where phase uncertainty depends on signal-to-noise ratio. Interestingly, the basal unit has a much higher signal-to-noise ratio that ice immediately on top of it. We show here a figure of the return power at ApRES-LDC location to illustrate this point.**

[Figure]

**We agree with the reviewer that we should describe in more detail the ApRES uncertainty and highlight the fact that uncertainty, that inversely relates to signal-to-noise ratio, is lower in the basal unit than in the ice above it. This is unusual as signal-to-noise ratio typically decreases with depth. We are rewriting the document to incorporate these points.**
**New line 226**
*We follow the standard ApRES uncertainty estimation technique described in Brennan et al. (2014) and Nicholls et al. (2015), where phase uncertainty depends on signal-to-noise ratio. The uncertainty inversely relates to signal-to-noise ratio and is lower in the basal unit than in the ice above it which is unusual as it typically decreases with depth.*

L263-264 what about less-simple explanations/alternative avenues?

**We are sure that less simple explanations can be elaborated, but we cannot think of an alternative that is consistent with the modelling results and the characteristics of the area.**

Fig9. It seems there are spots where the model suggests age jumps from around 1Ma to 2 Ma basically within one radar data point (e.g. at 75.5S and 125.3E). Is this an artefact or what is leading to these drastic differences?

**Former Fig. 9 (now Fig.11). There are a couple of reasons why this occurs in the HiCARS dataset. One issue is with the radar isochrones which have some discontinuities in locations like the one mentioned by the reviewer. This is due to the lower resolution radar and the ice melting so the deeper isochrones are harder/impossible to trace. Having fewer isochrones to constrain the model at these locations can lead to discontinuous results. Linked to this, the second issue is that further from the dome, there is more horizontal flow. This means that the 1D model is less appropriate for these locations. We can see from the reliability index (formerly Fig. 12, now Fig. 6) that further from the dome, eg. over Concordia trench, the model is unreliable. We discuss this in more detail in sec 5.1. Modelling limitations. Since over the main areas of study – LDC and NP – are not subject to these issues, we leave the discussion there.**

**New line 415:**

*However, where ice flows along the Concordia trench as seen in the HiCARS radar dataset, the model becomes less relibale (Fig. 6a). HiCARS uses a lower resolution radar system than the other datasets, and regions with ice flow can make IRHs untraceable. Over Concordia trench there appear to be some anomolies in the modelled maximum age (Fig. 13a) which are due to the discontinuities in the traced IRHs.*

Fig10 how is an age density at 1.2 Ma defined so close to EDC where there is no ice which is 1.2 Ma old (I am comparing to oldest ice near EDC in table III)?

**Former Fig 10a (now Fig 12a) uses the HiCARS dataset which covers a larger area but is also lower resolution than the other radar surveys. Table III uses higher resolution radar (DELORES and LDC-VHF) and radar lines which pass within 500m of EDC, closer than 2 km for the HiCARS radar dataset.**

L321 would be overestimated by how much? Is it possible to give rough estimates for reasonable potential SMB increases (e.g. surmised from existing modelling pre-MPT time slices)?

**We now give an estimate for this.**

**New line 372:**

*The oldest ice found at EDC was 800 ka, therefore information on the accumulation variation r(t) is limited to ice younger than this. Inferring accumulation variations from marine sediment cores (Lisiecki and Raymo, 2005) for ice between 1500-800 ka, we find that the average accumulation rate is around 6% higher during this time period than <800 ka. This would correspond to an under-estimation of layer thickness which equates to ~40 kyr over-estimation in the modelled age of ice between 1500-800 ka. This over-estimation is well within our uncertainty ranges for this age of ice*

L323 by how much would that few percent change in the thinning function translate into age changes?

**The thinning function uses an average thickness therefore variations of thicknesses over time are likely to cancel out or cause a small change of a few percent, which would result in an even smaller change in the inferred age.**

**New line 378:**
*this might only affect the thinning function and therefore inferred age by a few percent*

L325 "There are a few possible explanations for this" this sentence is not necessary. Suggest to provide possible explanations right away. -> 'this could be due to …'
**Suggestion accepted**

L327 if sliding would occur at the boundary between stagnant ice and above is there a way to detect this in the ApRES data? I am aware that ApRES is used for vertical velocities and not horizontal and I am not a radar expert, so please forgive this somewhat naïve question. There is a publication by Summers et al., 2021 which seems to suggest the extraction of horizontal velocities from ApRES.
**ApRES vertical velocity shows that, within the uncertainty of the data, the basal unit is vertically stagnant. This does not rule out the possibility of horizontal advection. Firstly, ice incompressibility will imply that the combination of both horizontal strain-rates is null, but basal ice could be extending in one horizontal direction and compressing in the other one. Horizontal velocity can be derived from vertical velocity only in simple scenarios. Secondly, null vertical velocity is not incompatible with spatially uniform horizonal velocity. However, our age model, with the reliability index, and the characteristics of the area - slow flow, relatively shallow thickness with no sign of subglacial melting- suggest a small role from horizontal advection in this particular location.**

**In any case it is interesting to clarify in the paper that ApRES is only indicating stagnant ice in the vertical.**
**New line 470:**
*The ApRES measurements indicate that ice is vertically stagnant however, that does not rule out the possibility of horizontal advection. Our age model, with the reliability index, and the characteristics of the area - slow flow, relatively shallow thickness with no sign of subglacial melting- suggest that horizontal advection has a small role in this particular location.*

L341 again, I don't think the age uncertainty is that high for the oldest ice in the EDC ice core. See comment earlier, maybe a typo which reoccurred here.
In the supplements of Bazin et al. 2013 std is given (801.5ka pm 9.6ka).
**Yes, corrected**

L345 check formatting of citation.
**We format the citation as follows *Obase et al. (2022, see Fig. 6 of that publication)***

L349 To me this seems to be a considerable limitation. How do you choose the cutoff depth where you don't trust the exponential age-depth profile anymore? If I look ~300 m above bedrock in your figure 11 true age could be anything between ca. 700 ka and 1200 ka depending on your cutoff depth. I think this should be discussed in more depth to give an idea how you quantify your uncertainties.
**We have tried to be as detailed and open as possible in this discussion and have now added more cautionary statements. Unfortunately, we only have 2 ice core profile in central east Antarctica – EDC and Dome Fuji. Both of which had basal melting so we are really in unexplored territory with LDC. And**

**as mentioned in the general comment, we require a more complex stagnant ice model to quantify our model uncertainties.**

L355 unclear what you mean by "seem to be well adapted"?
**New line 419:**
***therefore, our assumptions seem to be appropriate for areas close to a dome.***

L415-417 this is a very interesting notion. Right now basal drag in large scale 3D modelling exercises is either formulated by heuristics or established via inversion. However, if these methods overestimate basal drag they would have to be re-tuned to match present day observations and proxy reconstructions (i.e. ice would have to deform/flow less easily). So, I am not sure whether this would necessarily lead to increased sea level rise in the future. I suggest thus to rephrase the sentence into something more cautious.
**New L490 changed to *"If it is widespread, perhaps the resultant decrease in basal drag could be incorporated into future sea level estimates."***

L425 as per your paragraph further above, you don't know at which depth you cannot trust the exponential age-depth profile anymore and therefore ice could be much younger. I suggest you include this cautionary statement again here.
**Sentence added to new line 499 *"It is also unclear to what depth we can trust the exponential age-depth profile predicted by the Lliboutry assumption (Eq. 3)."***

L437 I suggest to drop the 'etc.' here.
**Suggestion accepted**

Some suggesions for the figures:

Figure 3. I am confused by the colormaps. Panel a and d show RdBu cmaps but only uniform blue, grey and red cmaps are shown in the colorbar. I assume the red and blue are combined in the figures. Please consider merging the colorbars to avoid confusion (you can e.g. define the melt rate as negative, colorbar would go from -3 – 250 m with a zero intercept in white).
**Both red and blue colour bars apply to all 4 panels. There is no melting in panels b and c, hence no red. Due to the inverted bedrock depth aspect of the model, there can be either melting (red) or stagnant ice (blue) at any given location, which is why the 2 colour bars apply to all panels. We are hesitant to merge the colour bars as they are showing 2 different quantities, it could be confusing to have different units for each half of a single joined diverging RdBu colour bar. However, we have rearranged them so melting is on the left going from 3 to 0 melt rate. Then on the right, we put the stagnant ice thickness from 0 to 250m. Therefore 0 (white in both cases) is towards the centre of the figure which makes it easier to understand the relationship but the colour bars are separate so it is still clear that the 2 colours are measuring 2 different quantities.**

You could also consider a diverging cmap for bedrock topography which makes it easier to identify mountains and valleys, but that's a question of taste. However, you could reduce the range for the greyscale colormap so bedrock features pop up more prominently, otherwise this is basically just a light gray background in the zoomed-in panels.

**We considered a diverging cmap for bedrock however, with the colour scales used for the radar lines, we find its becomes difficult distinguish between the colours of modelling results and the bedrock. So we keep the bedrock greyscale but have reduced the range to make features more prominent as suggested.**

Also the contour lines are somewhat busy/distracting. Maybe use fewer or skip completely.
**We find the surface contour lines useful to understand the location as they show the primary Dome Cand the secondary dome LDC. However, we agree that perhaps there was too much detail so we have now smoothed the contour lines so they are less distracting.**

Figure 6. x-axis should be unitless I assume.
**Yes, axis now reads *"Normalised vertical velocity"***

---

## Author Response (AR2)

**Author response**

Minor changes since 13/06/23 submission:

Line 5: There was a typo, "difference" has now been corrected to "different"

Eq. 5: variables $\chi^{iso}$ , $\chi^{mod}$ , $n^{iso}$ and $\sigma^{iso}$ had subscript labels but have now been changed to superscript to keep consistent with Eq. 4

Line 549: Beyond EPICA publication number added